# Anxiolytic and Antidepressant Effects of Organic Polysulfide, Dimethyl Trisulfide Are Partly Mediated by the Transient Receptor Potential Ankyrin 1 Ion Channel in Mice

**DOI:** 10.3390/pharmaceutics17060781

**Published:** 2025-06-14

**Authors:** Kitti Göntér, Viktória Kormos, Erika Pintér, Gábor Pozsgai

**Affiliations:** 1Department of Pharmacology and Pharmacotherapy, Medical School, University of Pécs, H-7624 Pécs, Hungary; viktoria.kormos@aok.pte.hu (V.K.); erika.pinter@aok.pte.hu (E.P.); 2Department of Pharmacology, Faculty of Pharmacy, University of Pécs, H-7624 Pécs, Hungary; pozsgai.gabor@gytk.pte.hu

**Keywords:** dimethyl trisulfide, mechanism, TRPA1, ion channels, depression, anxiety, chronic stress

## Abstract

**Background/Objectives**: Dimethyl trisulfide (DMTS) is a naturally occurring polysulfide with known antioxidant and neuroprotective properties. DMTS is a lipophilic transient receptor potential ankyrin 1 (TRPA1) ligand that reaches the central nervous system (CNS). Its role in the CNS, particularly regarding depression-like behaviour, has yet to be explored. This study investigates the influence of DMTS on stress responses and whether this effect is mediated through the TRPA1 ion channel, known for its role in stress adaptation. Using a mouse model involving three-week exposure, we examined the impact of DMTS on depression-like behaviour and anxiety and identified the involved brain regions. **Methods**: Our methods involved testing both *Trpa1*-wild-type and gene-knockout mice under CUMS conditions and DMTS treatment. DMTS was administered intraperitoneally at a dose of 30 mg/kg on days 16 and 20 of the 21-day CUMS protocol—in hourly injections seven times to ensure sustained exposure. Various behavioural assessments—including the open field, marble burying, tail suspension, forced swim, and sucrose preference tests—were performed to evaluate anxiety and depression-like behaviour. Additionally, we measured body weight changes and the relative weights of the thymus and adrenal glands, while serum levels of corticosterone and adrenocorticotropic hormone were quantified via ELISA. FOSB (FBJ murine osteosarcoma viral oncogene homolog B) immunohistochemistry was utilised to assess chronic neuronal activation in stress-relevant brain areas. **Results**: Results showed that CUMS induces depression-like behaviour, with the response being modulated by the TRPA1 status and that DMTS treatment significantly reduced these effects when TRPA1 channels were functional. DMTS also mitigated thymus involution due to hypothalamic–pituitary–adrenal (HPA) axis dysregulation. **Conclusions**: Overall, DMTS appears to relieve depressive and anxiety symptoms through TRPA1-mediated pathways, suggesting its potential as a dietary supplement or adjunct therapy for depression and anxiety.

## 1. Introduction

The expression of the transient receptor potential ankyrin 1 (TRPA1) ion channel is well known in peripheral nociceptive nerves, but there is increasing evidence of its central nervous system (CNS) presence [1]. *Trpa1* mRNA expression was confirmed in certain stress-related limbic brain areas, including the olfactory bulb, piriform cortex, and hypothalamus; however, the highest level of *Trpa1* mRNA was detected in the urocortinergic cells of the centrally projecting Edinger–Westphal nucleus (EWcp), both in mice and humans [2]. To confirm TRPA1 functionality, whole-cell patch-clamp recordings were performed on acute EWcp slices following the administration of JT010, a highly selective and potent TRPA1 agonist [3]. Researchers from our institution confirmed that *Trpa1* was downregulated in the EWcp in chronic unpredictable mild stress (CUMS)-model in mice and human suicide victims [2]. Moreover, altered stress adaptation ability was observed in *Trpa1*-gene-deficient mice in a single prolonged stress model of posttraumatic stress disorder (PTSD) [4]. This suggests that TRPA1 in EWcp neurons might contribute to regulating depression-like behaviour and the stress adaptation response in mice [2].

With regard to the role of TRPA1 within the central nervous system in depression and anxiety models, accumulating evidence suggests that TRPA1 channels exert a tonic facilitatory effect on affective behaviour, as the modulation of TRPA1 results in anxiolytic- and antidepressant-like effects in mice [5]. Chronic stress exposure was associated with upregulated TRPA1 expression in the hypothalamus, and behavioural improvements were observed following TRPA1 inhibition, indicating that TRPA1 may contribute to the maladaptive neuroendocrine and emotional responses elicited by prolonged stress [6]. These findings support a neuromodulatory function for TRPA1 in mood regulation, likely involving brain areas such as the periaqueductal grey and hypothalamus.

In addition to electrophilic and non-electrophilic activators, naturally occurring compounds can also be included as activators of TRPA1 ion channels. These include wasabi, cinnamaldehyde, eugenol, phenylpropenes of cloves, piperine in black pepper, curcumin or gingerol. The activation of TRPA1 by compounds produced by Brassica (mustard) and Allium (onions and garlic) plants, such as pungent isothiocyanates and allicin, respectively, is fascinating [7].

Among these agonists, lipophilic dimethyl trisulphide (DMTS) has been proven to reach the brain. DMTS is a sulphur-based molecule found in garlic, onion, broccoli, and similar plants. DMTS is readily available in high purity, is naturally occurring and stable, and can be reasonably priced. DMTS has an elimination half-life of 36 min and accumulates mainly in red blood cells [8]. It readily crosses the blood–brain barrier and can be detected in brain tissue, including stress-relevant brain areas, such as the EWcp [9,10].

We selected DMTS over other TRPA1 agonists and polysulfides based on its favourable pharmacokinetic characteristics, including blood–brain barrier penetration and central accumulation, as well as its chemical stability and commercial availability. Importantly, unlike many other TRPA1 ligands, DMTS has been shown to exert central effects without impairing locomotor or respiratory functions, making it a suitable and translationally relevant candidate for investigating TRPA1-mediated mechanisms in stress-related behaviour [11].

The literature suggests that sulphide attenuates depression-like behaviour induced by CUMS [12,13]. Some suggest that the effects of sulphides are mediated by their oxidised forms, called polysulfides [14,15,16], which have been detected in the CNS of rodents. DMTS is an organic polysulfide, and the main differences between organic and inorganic polysulfides are reflected in the half-life, purity, and availability. Organic polysulfides have more favourable pharmacokinetic properties. Despite growing interest in the role of sulphides within the central nervous system, their influence on anxiety and depression remains largely unexplored [17]. In our previous experiments, DMTS inhibited spontaneous motor activity and respiration in mice [18]. Given that central nervous system depressants such as benzodiazepines and barbiturates typically alleviate anxiety, we proposed that a carefully selected dose of DMTS could exert anxiolytic and antidepressant effects without adversely affecting locomotion or respiratory function. Our previous findings suggest that one of the main targets of DMTS effects is the TRPA1 cation channel [19]. The TRPA1 ion channel is activated by sodium polysulfide in the CNS astrocytes [20]. Our results imply that TRPA1 expressed on Chinese hamster ovary (CHO) cells is activated by DMTS and sodium polysulfide (POLY) [11,18]. Inhibition of the motor activity of mice by DMTS is mediated by the TRPA1 channel [18].

It raises the importance of research on the topic that the detailed pathomechanism of these disorders remains unclear due to the lack of a universally accepted animal model in basic research and that the effective treatment of anxiety and depression is also unmet with current treatment strategies [21].

In the present study, we aimed to investigate whether the TRPA1 agonist DMTS modulates stress-related brain processes, such as anxiety or depression. We tested the impact of DMTS on a mouse model of depression with special focus on the involvement of the TRPA1 channel. We carried out behavioural studies using *Trpa1* wild-type (WT) and knockout (KO) mice in a CUMS model to detect anxiety- and depression-like behaviour. We tested physical and endocrine parameters to assess the stress adaptation responses. FOSB immunohistochemical analysis was utilised as a marker of chronic neuronal activation and an enzyme-linked immunosorbent assay (ELISA) was performed to measure the plasma corticosterone (CORT) and adrenocorticotropic hormone (ACTH) levels.

This study builds upon our previously published findings [22] regarding the effects of DMTS in an acute stress model, where the involvement of the TRPA1 ion channel was demonstrated. In contrast to the acute paradigm, the present work utilises the CUMS model of depression, which is considered more translationally relevant for modelling human affective disorders due to its prolonged nature and closer alignment with the pathophysiology of chronic stress-related psychiatric conditions.

Overall, our primary hypothesis was that DMTS alleviates anxiety- and depression-like behaviour induced by chronic stress in mice. As a secondary hypothesis, we proposed that these effects are at least partially mediated by the TRPA1 ion channel.

## 2. Materials and Methods

### 2.1. Animals

Experiments were conducted on genetically modified male mice (adult, 25–30 g) lacking functional *Trpa1* and their WT counterparts. Age-matched animals were used. The animals were housed in a temperature- and humidity-controlled environment with a 12 h light–dark cycle (lights were turned on at 6:00 a.m.). Mice were housed in standard polycarbonate cages (365 mm × 207 mm × 144 mm) in groups of 4–6 mice per cage at the animal facility of the Department of Pharmacology and Pharmacotherapy, University of Pécs, Pécs, Hungary. They were provided standard rodent food and tap water ad libitum, ensuring their well-being. All experimental procedures were carried out according to the European Communities Council Directive of 2010/63/EU. The studies were approved by the Animal Welfare Committee of the University of Pécs and the National Scientific Ethics Committee for Animal Experiments in Hungary (permission number: BA/73/0476-9/2022). As previously described, *Trpa1* KO mice were bred using a C57BI/6 background [11].

### 2.2. Chemicals

DMTS (Sigma-Aldrich, Budapest, Hungary) solutions were prepared in physiological saline containing polysorbate 80 (Sigma-Aldrich, Budapest, Hungary). A 3% m/v solution was first prepared by dissolving polysorbate 80 in physiological saline. The DMTS stock was then formulated at a concentration of 10 mg/mL. This was subsequently diluted with saline to yield the final working concentrations. All formulations were administered intraperitoneally (i.p.) at a volume of 10 mL/kg. The vehicle employed in the experiments contained 1.5% m/v polysorbate 80.

DMTS exhibits rapid systemic distribution following parenteral administration, with an elimination half-life of 36 min in the rat. A recent preliminary study identified a final elimination half-life of 10.5 h [8,23,24,25]. It was demonstrated that DMTS is capable of crossing the blood–brain barrier both in vitro and in vivo, reaching the brain within 10 min of administration [10]. The short systemic exposure of DMTS necessitates repeated dosing when aiming for a sustained CNS concentration. In the absence of prior CNS-targeted pharmacological data for DMTS, our dosing regimen was empirically optimised based on behavioural efficacy and the absence of sedation. The chosen 30 mg/kg dose was identified through dose-reduction studies to represent the highest non-sedative concentration suitable for behavioural paradigms. The administration schedule (7 times every 60 min) was based on our previous work [11]. We were investigating the anti-inflammatory effect of DMTS and were looking for a way to administer relatively large doses of DMTS without compromising the behaviour of mice or eliciting toxic effects. The cumulative dose used in the present study (7 × 30 mg/kg, i.p.) is in the same magnitude as the one used in the rat pharmacokinetics study (200 mg/kg, i.m.) not counting the unknown first-pass metabolism [10]. Behavioural testing was performed 24 h after repeated DMTS administration. The concentration in the brain was most probably very small by then, but behavioural effects proved to be lasting. While the short half-life of DMTS imposes some limitations, our results indicate that repeated i.p. dosing ensures reproducible central effects, thus rendering the pharmacokinetic profile compatible with the present experimental aims.

### 2.3. Experimental Design

#### 2.3.1. Dose Determination and Treatment Groups

The appropriate intraperitoneal dose of DMTS was determined using OFT (see Section 2.5.1). The test was used to monitor the general locomotor activity of the *Trpa1* WT animals. An evaluation of behavioural tests for chronic stress can be compromised based on altered motor activity. The distance travelled by the mice and the time spent moving were evaluated. For dose testing, animals were divided into 4 groups. DMTS was evaluated at doses of 30 mg/kg and 40 mg/kg, alongside corresponding volumes of the vehicle control. In addition, there were naïve and vehicle-treated groups with 8 animals in each group.

After dose determination, in the second experiment, we divided the animals into 12 treatment groups. *Trpa1* KO mice were utilised to explore the mechanism of action and to validate the role of the TRPA1 receptor. We used stressed and non-stressed groups within the two mouse genotypes. Naïve, vehicle-treated, and DMTS-treated subgroups were included within WT and KO groups (8–16 mice per group; Table 1).

Animals were randomly allocated to experimental groups to minimise selection bias. All animals were identified solely by numerical codes marked on their tails, without an indication of the treatment, stress condition, or genotype thereby ensuring blinding during behavioural assessments.

#### 2.3.2. Experimental Schedule

The total experimental period lasted for 24 days. The CUMS paradigm was applied from day 0 to day 21. Treatments were applied on days 16 and 20 (Figure 1A). There was an interval of 3 days between the two treatment days to allow for DMTS elimination from the body. DMTS and its vehicle were administered i.p. with a 30 G needle in the lateral lower abdomen of the conscious animal. Injections were given seven times, once every hour at a dose of 30 mg/kg. Behavioural tests were carried out 24 h after the last injection. The animals were terminated on day 23, 48 h after the last stressor was applied, and their organs were collected. Control animals were measured in behavioural tests on days 7 and 11 and were treated with DMTS or the corresponding vehicle on the preceding days, in the same way as stressed animals (Figure 1B).

### 2.4. CUMS Paradigm

We used the CUMS model of depression, because the validity of this model was repeatedly confirmed [12,26,27]. The 3-week chronic stress paradigm consisted of 4 different mid-day stressors (restraint stress, tilted cage, shaker stress, dark room), applied between 10 a.m. and 2 p.m., as well as 3 types of overnight stressors (wet bedding, social isolation, group holding) (Table 2). In the case of ”restraint stress”, the mice were closed into a plastic tube equipped with a perforated conical tip and several additional ventilation holes for 60 min. For the “tilted cage”, their cages were fixed at a 45° angle for 3 h. For the “shaker stress”, mice were placed on a laboratory shaker set to 60 rounds per minute. For the “dark room”, animals were kept in the dark for 3 h during their daytime light hours. The overnight stress-exposure period started at 6 p.m. and lasted until 6 a.m., except on “group holding” days, where animals were left undisturbed. For “wet bedding”, the sawdust was moisturized with 250 mL of tap water. The next morning mice were placed on fresh, dry bedding. For “social isolation”, mice were individually housed overnight. The next morning, the original groups, previously housed in the same cage, were reunited. Stressors were applied randomly twice daily, day and night, for 3 weeks (Table 2). All groups of animals received the stressors in the same order, and no stressors were applied during the last 48 h. Control groups were not stressed. The body weights of all mice were measured on days 1 and 15. The last body weight measurement took place before the anaesthetic injection on day 24, immediately before perfusion or decapitation.

### 2.5. Behavioural Tests

Five well-established behavioural tests were used to verify the anxiety (open field test (OFT) and marble burying test (MBT)), the anhedonia (sucrose preference test (SPT)) and the depression-like behaviour (forced swim test (FST), and tail suspension test (TST)), as well as to test the impact of DMTS.

#### 2.5.1. Open Field Test

The OFT was conducted before performing the CUMS protocol to determine the optimal dosage of DMTS that does not impede the spontaneous movements of the animals. Animals were placed in the middle of a 60 × 60 cm box surrounded by 60 cm-high walls. The behaviour of the animals was filmed for 7 min with a digital camera, and the last 5 min of the video recordings were evaluated. The evaluation began 30 s after the appearance of the animal in the open arena. Doses of 30 mg/kg and 40 mg/kg of DMTS and respective doses of the vehicle were tested. The assessment continued for 5 min. Locomotor activity was assessed using Noldus EthoVision XT v15.0 software, which recorded both the duration of movement and the total distance travelled by each animal.

Later in the setting of the CUMS protocol, the OFT was used to assess the anxiety level and exploratory behaviour of the mice on day 21. In this case, we evaluated the time spent in the centre of the opened arena. Anxious animals spend less time in the open central part of the arena. The time spent in the peripheral area alongside the walls is proportional to the level of anxiety [28].

#### 2.5.2. Marble Burying Test

Mice were subjected to MBT for 30 min on day 17 of the CUMS protocol. Animals were placed one by one into a separate cage, where 24 coloured glass marbles of about 1 cm in diameter had been placed. After thirty minutes, the number of marbles buried by the animals in the box was recorded. Marbles buried up to at least two-thirds of the diameter were counted as buried. More buried marbles indicate a higher level of anxiety [29,30].

#### 2.5.3. Sucrose Preference Test

On day 21 of the CUMS protocol, mice were subjected to SPT for 12 h. Anhedonia is an important sign of depression characterised by a lack of interest in activities that normally bring pleasure. In animals, this can be assessed through the SPT. The test involves habituating the mice to two drinking bottles, one containing a 1% m/v sucrose solution and the other containing tap water. The two bottles were offered randomly over four nights without separating the animals. On the second day, the animals were deprived of water for 12 h before the dark phase of their circadian rhythm. Mice were placed individually into cages lined with clean litter and were randomly offered two drinking bottles, one with tap water and the other with sucrose solution at the beginning of the dark cycle. The consumption of tap water and sucrose solution was measured by weighing the bottles before and after the experiment. The sucrose preference was calculated as $$\frac{Consumption\;of\;sucrose\;solution}{\left(consumption\;of\;water+consumption\;of\;sucrose\;solution\right)}\times 100$$

#### 2.5.4. Forced Swim Test and Tail Suspension Test

On day 17 of the CUMS protocol, mice were subjected to FST. Animals were placed in a clear plastic container partially filled with water at 23 °C for 6 min. This test was conducted avoiding danger and also the escape of mice from the container. Animals were recorded, and the last 4 min of the test was evaluated. Water was drained from the fur of mice and they were returned to their cages when dry.

On day 21 of the CUMS protocol, mice were subjected to TST for 5 min. The TST involved holding the animals by their tails without causing any pain and securing them in a suspended position using adhesive tape. The suspension surface was 50 cm above the ground. The behaviour of the mice was recorded using a video camera. Care was taken not to injure the animals.

In both tests, EthoVision XT v15.0 (Noldus) software was used to evaluate the time the animals spent moving and immobile, as well as the frequency of high-activity periods.

Experiments were conducted in a secluded room to ensure that the animals did not witness each other swimming or hanging. Bottles used in the experiments were thoroughly cleaned after each use. Mice were allowed to acclimate to the laboratory for 60 min each day for the three days before the experiment.

### 2.6. Termination

#### 2.6.1. Perfusion and Tissue Collection

Forty-eight hours after any manipulation, half of each experimental group was anaesthetised via urethane (i.p., 2.4 g/kg). Animals were weighed, and their tails were clipped. Mice were then perfused transcardially with 20 mL of ice-cold 0.1 M phosphate-buffered saline (PBS; pH 7.4), followed by 150 mL of 4% paraformaldehyde solution in Millonig buffer (pH 7.4) for 15 min. The adrenal glands and thymus were removed and weighed using a Entris224i-1S Sartorius analytic scale (Sartorius AG, Göttingen, Germany). Data were corrected for body weight.

Brain samples were dissected and post-fixed in the same fixative solution for 72 h at 4 °C. We collected 5 series of 30 μm sections (Leica VT1000S vibratome; Leica Biosystems, Wetzlar, Germany) and stored them in PBS containing 0.01% sodium azide at 4°C. Immunohistochemistry was performed to detect the expression of FOSB protein in the EWcp (Bregma −2.92 mm to −4.04 mm), dorsal raphe nucleus (DR; Bregma: −4.04 to −4.16 mm), basolateral amygdala (BLA; Bregma: −0.58 to −1.46 mm), lateral septum (LS; Bregma: 0.26 to −0.10 mm), paraventricular nucleus of the thalamus (PVT; Bregma: −0.22 to −0.70 mm), paraventricular nucleus of the hypothalamus (PVN; Bregma: 0.26 mm), bed nucleus of the stria terminalis (BST; Bregma: 0.62 to −0.22 mm) and periaqueductal grey matter (PAG; Bregma: −3.80 to −4.80 mm).

#### 2.6.2. Decapitation and Blood Sample Collection

The other half of each experimental group was terminated through cervical dislocation, and blood was collected (500 μL) via cardiac puncture into a syringe containing 50 μL of a 7.5% m/m EDTA solution (Sigma-Aldrich, St. Louis, MO, USA). Plasma samples were stored at −80 °C until the ACTH and CORT content was determined via ELISA.

### 2.7. FOSB Immunohistology

Immunohistochemistry was carried out as described previously by Kormos et al. [26]. After washing, sections were treated with 1% hydrogen-peroxide (H_2_O_2_; Sigma-Aldrich, St. Louis, MO, USA) in PBS to inhibit the endogenous peroxidase activity. Permeabilisation was carried out with a 0.5% Triton X-100 solution for 30 min. Normal horse serum (2%; NHS, Jackson Immunoresearch, Europe Ltd., Suffolk, UK) in PBS for 30 min was applied to block the non-specific binding sites. Subsequently, sections were transferred into a solution of mouse anti-FOSB antibodies (Santa Cruz, sc-48 Santa Cruz Biotechnology Inc., Santa Cruz, CA, USA) at a 1:500 dilution in PBS and 2% NHS in PBS overnight, at room temperature. After 2 × 15 min of washing in PBS, sections were treated with 1:200-diluted biotinylated horse anti-mouse antibodies for 60 min (Vectastain ABC Elite Kit, Vector Lbs., Burlingame, CA, USA) followed by 2 × 15 min rinses in PBS. Preparations were then treated with a peroxidase-conjugated avidin–biotin complex (Vectastain ABC Elite Kit; Vector Laboratories, Newark, CA, USA) according to the supplier’s protocol for 60 min. After rinsing in PBS, the immunoreaction was developed in Tris buffer (pH 7.4) with 0.02% 3,3′ diamino-benzidine (DAB) (Sigma-Aldrich, St. Louis, MO, USA ) and 0.03% *v*/*v* H_2_O_2_. The chemical reaction was accomplished under visual control using a microscope to optimise the signal/background ratio and was stopped after 7 min with PBS. Afterwards, the preparations were rinsed with PBS and mounted on slides coated with gelatine. After drying, slides were transferred into ascending ethanol solutions (70% (10 min), 96% (10 min), absolute (10 min)), and then into xylene for 2 × 20 min and coverslipped using Depex (Fluka, Heidelberg, Germany) mounting medium.

### 2.8. Microscopy and Digital Imaging

DAB-labelled sections were examined and digitalised using a Nikon Microphot FXA microscope with a Spot RT camera (Nikon, Tokyo, Japan). Five serial sections were photographed at a 90 lm luminous flux for each animal. The areas examined were the EWcp, DR, BLA, LS, PVT, PVN, BST and PAG.

### 2.9. Enzyme-Linked Immunosorbent Assay

Serum ACTH (VTSZ: 38220000) and CORT (VTSZ: 38220000) concentrations were measured using a commercially available ELISA set (Abcam, Cambridge, UK). Blood was collected from the animals and centrifuged (3000 rpm, 5 min at 4 °C), and the supernatant was isolated and stored at −80 °C until assayed.

When determining CORT, 25 μL of sample and standard solutions was added to the pre-coated antibody plate and incubated for 2 h at 37 °C followed by washing with 200 μL of wash buffer. Then, 50 μL of the SP Conjugate reagent was added and incubated for 30 min with shaking. After another wash, 50 μL of the chromogen substrate was added to each well and incubated until the optimal blue colour density developed. The reaction was stopped by adding 50 μL of stop solution, and the absorbance was read at 450 nm using a microplate reader (Multiskan EX, Thermoscientific, Waltham, MA, USA).

During ACTH determinations, 50 μL of standard or sample and 50 μL of antibody cocktail were added to the wells. After incubation for 1 h at room temperature, 350 μL of wash buffer per well was added, followed by 100 μL of 3,3′,5,5′ tetramethylbenzidine solution. The reaction was terminated with 100 μL of stop solution per well after incubation for 20 min. Optical density (OD) values were read at 450 nm.

The serum CORT concentration was measured in ng/mL, and the ACTH concentration was in pg/mL. In both cases, standard curves were generated, and the unknown sample concentrations were calculated.

### 2.10. Evaluation Methods

#### 2.10.1. Noldus EthoVision XT 15

The Noldus EthoVision XT system enables the objective tracking of animal behaviour under laboratory conditions. Employing an overhead camera and contrast-based detection, it offers a dependable alternative to observer-dependent methods. The platform produces a broad spectrum of quantitative data and ensures reproducibility across experimental sites [31].

The OFT evaluation starts 30 s after the animal appears in the arena and lasts for 5 min. The time spent moving and the distance travelled during this time are assessed. In the case of FST, the last 4 min of the 6 min recording was evaluated. We measured how long the animal was inactive and how often it entered a higher activity state based on the changes in the outline of the image of the mouse. Similarly, in the case of TST, the program assessed the time spent immobile and the number of active periods.

#### 2.10.2. ImageJ

Quantification of FOSB immunoreactivity was performed using ImageJ software (Version 1.54d). By delineating the region of interest and applying defined threshold parameters, the software distinguishes cells from background based on the optical density, quantifies cell numbers within the selected area, and determines its surface area [32].

### 2.11. Statistics

Data are presented as the mean and standard error of the mean. Three-way analysis of variance (ANOVA) was used to evaluate the immunohistochemistry data. The normality of the data distribution was checked based on the Shapiro–Wilk test and the homogeneity of variance was assessed via Bartlett’s Chi-squared test [33]. Behavioural data were evaluated via one- (OFT for dose determination) and three-way ANOVA. All post hoc analyses were performed using the Tukey test based on the ANOVA’s first- or second-order effects. Statistical evaluation was performed using GraphPad Prism v8.0 software (GraphPad Software, San Diego, CA, USA).

#### Hypothesis Verification

Main effects

Null hypothesis 1 (NH1): The measured output is the same in stressed and non-stressed animals.

Null hypothesis 2 (NH2): The measured output is the same in *Trpa1* WT and KO animals.

Null hypothesis 3 (NH3): The measured output is the same in naïve, vehicle-treated, and DMTS-treated animals.

Two-way interactions

Null hypothesis 4 (NH4): Pooling stressed and non-stressed animals, the impact of the *Trpa1* genotype (*Trpa1* WT vs. KO) is the same in naïve, vehicle-treated, and DMTS-treated animals.

Null hypothesis 5 (NH5): Pooling *Trpa1* WT and KO mice, the effect of being naïve, vehicle-treated, and DMTS-treated is the same in stressed and non-stressed animals.

Null hypothesis 6 (NH6): Pooling naïve, vehicle-treated, and DMTS-treated mice, the effect of the *Trpa1* genotype (*Trpa1* WT vs. KO) is the same in stressed and non-stressed animals.

Three-way interaction

Null hypothesis 7 (NH7): There is no three-way interaction among the effects of chronic stress, *Trpa1* genotype, and treatment.

Referring to NH1–7, we later indicate the rejection of the respective null hypotheses within the manuscript.

## 3. Results

### 3.1. Finding the Suitable Dose of DMTS Using Open Field Tests

The distance travelled was measured in centimetres (Figure 2A), and the time spent moving was measured in seconds (Figure 2B). DMTS at 40 mg/kg significantly reduced both the time spent moving and the distance travelled in the observed time interval compared to the untreated group. No inhibition was observed at 30 mg/kg of DMTS treatment. This dose was used in the following experiments using CNS models of depression-like behaviour. Neither the vehicle of 40 mg/kg nor that of 30 mg/kg DMTS affected the spontaneous movements of the animals. The vehicle of 30 mg/kg DMTS was used in further studies.

### 3.2. Validity of the Model

#### 3.2.1. Endocrine and Physical Parameters

##### Changes in Body Weights

In order to test the effects of CUMS, body weight changes in the animals were recorded (Figure 3A). Wild-type, non-stressed mice tended to gain weight over the three weeks. CUMS effectively reduced the body weight gain in non-injected, vehicle-treated and DMTS-treated *Trpa1* WT mice (*p* < 0.001) compared to non-stressed mice. In contrast, KO mice did not lose weight significantly. The main effects of stress (*p* < 0.0001; NH1) and genotype (*p* < 0.0002; NH2) were significant, while the treatment effect was below the statistical threshold. The stress × genotype (*p* < 0.0004) interaction was significant (Table 3). DMTS treatment did not reverse the effect of CUMS on body weight changes in WT mice in comparison with naïve or vehicle-treated stress-exposed WTs. CUMS did not induce body weight changes in KO mice, as revealed by post hoc comparisons in non-injected, vehicle-treated, and DMTS-treated pairs of KO groups. Interestingly, non-stressed KO mice did not gain weight in any treatment groups, unlike non-stressed WT mice that gained 2–3 g (main genotype effect; *p* > 0.001). A comparison of body weight data (Figure 3B) before CUMS exposure revealed statistical differences between *Trpa1* WT and KO mice (main genotype effect; *p* = 0.004) and between the *Trpa1* WT naïve or vehicle-treated and DMTS-treated mice (*p* < 0.001). Post hoc tests confirmed that CUMS-exposed *Trpa1* WT mice had lower body weights than control ones (main stress effect; *p* < 0.0001). The absolute body weights in stressed WT and *Trpa1* KO mice did not differ significantly. All of the stress × genotype (*p* < 0.0001; NH6), stress × treatment (*p* = 0.0280; NH5), treatment × genotype (*p* = 0.0045; NH4), and treatment × stress × genotype (*p* = 0.0364; NH7) interactions were significant. Body weight data collectively prove that the stressors were effective in WT mice; interestingly, KO mice react differently to the same stress effect (Table 3).

##### Adrenal Gland and Thymus Weights

There was no difference in basal relative adrenal weights between the two genotypes. The relative adrenal weight was higher after exposure to CUMS (main effect of stress; *p* > 0.001; NH1) both in *Trpa1* WT and KO mice in all treatment groups and after treatment in *Trpa1* WT and KO, non-stressed mice (main effect of treatment; *p* < 0.0001; NH3) in comparison to the naïve and vehicle-treated groups, without a difference in genotypes (Figure 3C). There was no interaction (Table 3). There was no difference in the basal relative thymus weights between the two genotypes. CUMS exposure significantly reduced the thymus weight (main effect of stress; *p* = 0.0028; NH1) in WT but not in the *Trpa1* KO mice (main effect of genotype; *p* > 0.0001; NH2) in the naïve and vehicle-treated groups (Figure 3D). In WT mice that received DMTS treatment in addition to chronic stress, there was a lack of reduction in the thymus weight (*p* < 0.0078) compared to the naïve and vehicle-treated groups. There was an interaction between stress and genotype (*p* = 0.0017).

##### CORT and ACTH Serum Concentrations

There was no difference in basal CORT levels between the two genotypes. CUMS significantly enhanced the CORT concentrations (main effect of stress; *p* = 0.0357; NH1) only in WT mice, with a difference in genotypes (Figure 3E). DMTS treatment significantly boosted the CORT concentration in non-stressed animals without a difference in genotypes (main effect of treatment; *p* = 0.0493; NH3). There was no interaction among the three factors.

There was a significant difference in basal ACTH levels between the two genotypes (Figure 3F). The baseline ACTH level in *Trpa1* KO mice was slightly but significantly higher than in WT mice (*p* < 0.001; NH2). ANOVA revealed that CUMS influenced ACTH levels differently in the two genotypes (genotype × stress interaction; *p* < 0.0001; NH6). Post hoc tests showed that CUMS effectively increased the serum ACTH concentration in WT mice (main effect of stress; *p* < 0.0001; NH1); however, it failed to induce any further increase in KO mice. DMTS treatment significantly increased the serum ACTH concentrations (*p* = 0.0040), without a main effect, only in WT mice (treatment × genotype interaction; *p* = 0.0323; NH4; Table 3).

#### 3.2.2. Behavioural Tests

##### Tests to Detect Anxiety Level

Time spent on the periphery (Figure 4A) was one of the main anxiety parameters of the OFT, and there were no differences in the baseline between the two genotypes. Surprisingly, the chronic stress effect in any genotype did not affect this value.

However, DMTS treatment had different effects on the two genotypes (treatment × genotype interaction; *p* = 0.0118; NH4). DMTS treatment increased the time spent in the periphery in WT, non-stressed mice, while it had no effect in *Trpa1* KO, non-stressed animals. Nevertheless, DMTS treatment had no effect in WT, stressed animals. However, it reduced the time spent in the periphery zone in *Trpa1* KO, stress-exposed animals (treatment × stress interaction; *p* = 0.0002; NH5). We also examined the distance travelled at the periphery compared to the central zone (Figure 4B). There was also no difference in baseline, stress, and treatment-free values between the two genotypes. *Trpa1* WT and KO mice moved more in the periphery zone after CUMS exposure (main effect of stress; *p* < 0.0001; NH1), and this was also observed in non-stressed WT DMTS-treated animals (main effect of treatment; *p* = 0.0333; NH3). In contrast, in KOs (treatment × genotype interaction; *p* < 0.0001; NH4), the impact of DMTS was observed in the stressed animals (treatment × stress interaction; *p* < 0.0001; NH5), with less movement in the periphery zone after treatment (Table 3).

Regarding the marble burying test, we observed a baseline difference in anxiety levels between the two genotypes. *Trpa1* KO mice buried significantly more marbles than WTs (main effect of genotype; *p* < 0.0001; NH2). Chronic-stress-exposed WT animals buried significantly more marbles compared to the non-stressed animals in the MBT (main effect of stress; *p* > 0.0001; NH1; Figure 4C), but not the *Trpa1* KO groups (stress × genotype interaction; *p* > 0.0001; NH6). Concerning DMTS treatment, we observed that the effect of the substance resulted in an increase in the number of buried marbles in WT non-stressed animals but a decrease in stressed animals compared to naïve or vehicle-treated groups (treatment × stress interaction; *p* = 0.0003; NH5). Furthermore, the treatment × genotype × stress interaction was significant (*p* = 0.0047; NH7; Table 3).

##### Tests to Detect Anhedonia and Depression-like Behaviour

The preference of WT animals for sweetened water was significantly higher than that of *Trpa1* KO animals in all treatment groups in the SPT (main effect of genotype; *p* > 0.0001; NH2; Figure 4D). CUMS exposure affected the sucrose preference in WT and *Trpa1* KO animals (main stress effect; *p* > 0.0001; NH1). DMTS treatment increased the sucrose preference—i.e., reduced anhedonia—in stressed WT animals (main effect of treatment; *p* = 0.0474; NH3), with no impact in KO animals (stress × treatment × genotype interaction; *p* = 0.0359; NH7; Table 3).

FST and TST assessed the immobility time representing depression-like behaviour.

Regarding the forced swimming test, we found a baseline difference in depression levels between the two genotypes. Non-stressed *Trpa1* KO mice had higher immobility scores than their WT counterparts (main effect of genotype; *p* > 0.0001; NH2). CUMS affected the active duration (the main effect of stress; *p* < 0.0001; NH1) differently in the two genotypes in the FST (stress × genotype interaction; *p* > 0.0001; NH6; Figure 4E). *Trpa1* KO mice showed no change in activity after chronic stress. This augmented depression-like behaviour was not altered by CUMS exposure in *Trpa1* KO mice. DMTS treatment significantly enhanced the activity of WT, chronic stress-exposed animals (main effect of treatment; *p* = 0.0233; NH3). The treatment × genotype × stress interaction was also significant (*p* = 0.0070; NH7; Table 3).

In all treatment groups, *Trpa1* KO animals showed significantly lower activity than the WTs in the TST (main effect of genotype; *p* > 0.0001; NH2; Figure 4F). As a result of CUMS, WT mice had significantly lower activity (*p* > 0.001) in naïve and vehicle-treated groups. DMTS-treated, stressed animals produced a higher level of activity (main effect of treatment; *p* = 0.0045; NH3). The impact of CUMS was not observed in *Trpa1* KO mice (stress × genotype interaction; *p* = 0.0323; NH6), taking into account all treatment groups. The treatment × genotype × stress interaction was also significant (*p* = 0.0222; NH7; Table 3).

### 3.3. Pattern of FOSB Neuronal Activation in Stress-Related Brain Areas

The neuronal activation was assessed via FOSB immunocytochemistry in eight stress-related brain areas.

#### 3.3.1. Centrally Projecting Edinger–Westphal Nucleus

We found no difference in basal FOSB levels between the two genotypes in the EWcp. FOSB reactivity was higher in WT naïve and vehicle-treated groups following CUMS exposure (main effect of stress; *p* = 0.0008; NH1). ANOVA found the main effects of genotype (*p* = 0.0014; NH2) and stress × genotype (*p* < 0.0001; NH6) interactions to be significant (Figure 5). The DMTS-treated group showed significantly lower neuronal activation after CUMS exposure compared to the naïve or vehicle-treated groups (*p* < 0.0001; NH3), but only in the WT mice. The treatment × stress × genotype interaction was significant (*p* = 0.0448; NH7; Table 4).

#### 3.3.2. Dorsal Raphe Nucleus

We found no difference in basal FOSB levels between the two genotypes in the DR. In naïve and vehicle-treated WT mice, CUMS induced more than three-fold FOSB positivity (*p* = 0.0002). The effect of stress (*p* = 0.0002; NH1) and genotype (*p* = 0.0338; NH2), in addition to stress × treatment (*p* = 0.0276; NH5) and treatment × stress × genotype (*p* < 0.0057; NH7) interactions, had a statistical interaction (Table 4). In contrast, if WT mice were DMTS treated, the FOSB elevation was significantly reduced (*p* = 0.004; NH3). CUMS and DMTS treatment did not induce a rise in FOSB expression in the DR of KO mice (Figure 6).

#### 3.3.3. Periaqueductal Grey Matter

Basal FOSB reactivity did not differ between the two genotypes in the PAG brain area. CUMS influenced FOSB expression in the PAG (main effect of stress; NH1; *p* < 0.0001 Figure 7). Stressed WT mice showed almost three times higher neuronal activity than non-stressed animals. CUMS did not increase FOSB immunoreactivity in KO mice (stress × genotype interaction; *p* = 0.0006; NH6). DMTS treatment effectively reduced FOSB immunoreactivity in the PAG in WT stressed animals (main effect of treatment; *p* = 0.0311; NH3), but it had no effect in *Trpa1* KO ones (treatment × genotype interaction; *p* = 0.0128; NH4) or non-stressed WT animals (treatment × stress interaction; *p* = 0.0005; NH5). The treatment × stress × genotype interaction was not significant (*p* = 0.0555; Table 4).

#### 3.3.4. Paraventricular Nucleus of the Thalamus

We found a difference in basal FOSB reactivity between the two genotypes in the PVT brain area. KO, non-stressed mice showed higher FOSB-positive cell counts in the PVT than their WT counterparts (*p* = 0.0041; NH2). CUMS exposure significantly increased FOSB immunoreactivity in WT mice in the PVT (*p* = 0.0468; NH1; Figure 8) without the main effect of treatment but had no effect in *Trpa1* KO mice (stress × genotype interaction; *p* = 0.0001; NH6). DMTS treatment enhanced FOSB immunoreactivity in WT non-stressed animals, and it reduced FOSB-positive cell counts in WT stressed animals. Still, in both cases, the effect was only a tendency and not statistically significant. DMTS treatment did not affect KO mice. The treatment **×** stress **×** genotype interaction was significant (*p* = 0.0397; NH7; Table 4).

#### 3.3.5. Lateral Septal Nucleus

We found a difference in basal FOSB reactivity between the two genotypes in the LS brain area. *Trpa1* KO non-stressed animals exhibited significantly lower FOSB-positive cell counts than their WT counterparts (main effect of genotype; *p* > 0.0001; NH2). ANOVA revealed the main effect of stress in the LS (*p* = 0.0004; NH1; Figure 9). In WT mice, CUMS increased the FOSB-positive cell counts, but KO mice did not show CUMS-induced FOSB elevation, but only a tendency. DMTS treatment significantly reduced the number of FOSB-positive cells in WT stressed animals compared to the naïve or vehicle-treated groups (*p* = 0.0010), without the main effect of treatment. DMTS treatment did not impact *Trpa1* KO mice (treatment × genotype interaction; *p* = 0.0255; NH4). The treatment × stress × genotype interaction was also significant (*p* = 0.0296; NH7; Table 4).

#### 3.3.6. Paraventricular Nucleus of the Hypothalamus

For basal FOSB reactivity, no difference was found between the two genotypes in the PVN brain area. The assessment of FOSB immunoreactivity in the PVN of treated mice (Figure 10) revealed the main effects of stress (*p* < 0.0001; NH1) and genotype (*p* < 0.0005; NH2) to be highly significant. ANOVA found an interaction between treatment × stress (*p* = 0.0077; NH5) and treatment × genotype (*p* = 0.0257; NH4). Post hoc comparisons revealed that CUMS caused about twice the FOSB increase in naïve and vehicle-treated WT and KO animals (*p* < 0.01) without the main genotype effect. Interestingly, DMTS treatment caused an effective decrease in the number of FOSB-positive cells in WT chronic stress-exposed mice (*p* = 0.0001) but not in the KO stressed mice (treatment × genotype interaction; *p* = 0.0257; NH4). DMTS treatment did not affect non-stressed WT and KO mice (treatment × stress interaction; *p* = 0.0077; NH5; Table 4).

#### 3.3.7. Bed Nucleus of the Stria Terminalis

We found a difference in basal FOSB reactivity between the two genotypes in the BST (Figure 11). FOSB immunoreactivity was significantly higher in the WT mice compared to the non-stressed groups in every treatment group as a result of CUMS (main effect of stress; *p* < 0.0001; NH1), but chronic stress exposure did not affect *Trpa1* KO mice (stress × genotype interaction; *p* = 0.0158; NH6). In addition, the FOSB cell count was significantly higher in *Trpa1* KO mice in comparison to WT animals (main effect of genotype; *p* < 0.0001; NH2). There was no main effect of treatment or any other interaction (Table 4).

#### 3.3.8. Basolateral Amygdala

The influence of CUMS and treatment on FOSB neuronal activation in the BLA did not reach significance in any treatment groups (Figure 12). Treatment, stress, genotype, and interactions were also ineffective (Table 4).

## 4. Discussion

We tested the effects of a naturally occurring polysulfide found in garlic—DMTS—on chronic stress-induced anxiety and depression-like behaviour. We hypothesise that the TRPA1 ion channel might be involved as a possible mediator of DMTS effects. To the best of our knowledge, the anti-anxiety and anti-depressant effects of DMTS have not been studied before. However, a recent study by our research team has demonstrated the role of DMTS in acute stress [22].

### 4.1. Validity of the Model

We decided to utilise the CUMS model because it is a widely recognised and well-studied model in mice [34]. Exposure to CUMS resulted in typical physiological [35,36,37], hormonal (ACTH, CORT), and behavioural (anxiety, anhedonia, and depression-like) changes indicating heightened activity within the HPA axis [26,38]. The stress induced by the anaesthetic injection administered before perfusion may have resulted in acute elevations in ACTH and CORT levels. Despite this potential confounding factor, our observations of increased adrenal weight, diminished thymus mass, and body weight reduction strongly indicate prolonged constitutive activity of the HPA axis. These findings confirm the validity and reliability of our model. An interesting observation is that in the absence of the *Trpa1* gene, animals respond differently to the effects of chronic stress, suggesting a stress adaptation disorder with the lack of this gene [2].

### 4.2. Normal Stress Response Requires TRPA1 Ion Channel, but the Impact of DMTS Is Independent of TRPA1 Regarding Physical and Endocrine Parameters

We found TRPA1-related basal alterations in ACTH levels, body weight changes, and absolute body weights but not in the relative weights of the thymus and adrenal glands or CORT levels. Non-stressed *Trpa1* KO mice showed reduced body weight gain compared to WT ones, and their absolute body weight was lower, as well. This could be explained by the potential anorexigenic effect of TRPA1. Some studies suggest that the lack of TRPA1 leads to the upregulation of urocortin 1 (UCN1) [2]. UCN1 belongs to the corticotrophin-releasing hormone (CRH) family, with an anorexigenic effect [2,39]. Basal levels of ACTH were elevated in KO mice and could not increase further upon CUMS. No basal difference in CORT levels was found, but the CORT level response to chronic stress differs between the two genotypes. A statistically insignificant elevation of CORT levels was observed in response to stress in KO animals. This suggests that TRPA1 modulates the HPA axis response at the anterior pituitary or adrenal cortex levels. There was no basal difference in thymus weights between the two genotypes, but there was a difference in the response to chronic stress. No stress-induced alteration of the thymus weight was observed in gene KO animals. This correlates with increased CORT levels under chronic stress, as glucocorticoids exert an immunosuppressive effect associated with a decrease in the thymus weight. TRPA1 may modulate lymphocyte differentiation during chronic stress, as well. The TRPA1 protein has a dual role in immunity. It acts as a sensor for cellular stress, tissue injury, and harmful external stimuli and triggers defensive responses. However, when not properly regulated, it can also contribute to the impaired function of the immune system [40,41]. TRPA1 activity may play a role in developing immune cells of primary and secondary immune organs [41,42,43].

Overall, chronic stress affected all parameters in *Trpa1* WT animals. Most parameters were not affected by stress in KO mice, except the weight of the adrenal glands. The HPA axis function is altered in the absence of the TRPA1 ion channel, and the channel may be required for a normal stress response.

DMTS treatment resulted in increased resting CORT and ACTH levels, as well as adrenal gland weights. This was not observed regarding ACTH levels in KO animals. DMTS affected these values differently in stressed animals. In the early 2000s, Russo and colleagues proved that H_2_S could modulate the hypothalamic–pituitary system [44]. The effects of DMTS on CORT levels and adrenal weights are indeed not mediated through the TRPA1 ion channel. Elevated ACTH levels may have occurred due to the involvement of the TRPA1 ion channel. DMTS probably affected the pituitary gland via the TRPA1 ion channel and the adrenal gland through other routes. The reduction in the thymus weight under chronic stress was prevented by DMTS treatment, suggesting that it somehow diminished the immunosuppressant effect of glucocorticoids. The TRPA1 ion channel might be implicated, given the difference in KO animals. The protective effect of sulphides has been described in several papers explaining the positive effect of DMTS on the immune system under oxidative stress [45] and chronic endoplasmic reticulum stress [46]. Although there was no significant effect on body weight changes and the absolute body weight, there was a trend towards a slight reduction in body weight gains in WT animals. The TRPA1 ion channel might have anorexigenic effect [47].

Overall, DMTS treatment elevates CORT levels in both genotypes of control animals. DMTS treatment did not decrease the thymus size in wild-type mice and it did not prevent thymus weight reductions in KO ones because the weight did not increase in response to stress. Our results suggest that the effect of DMTS in the control is not TRPA1-dependent. DMTS has no impact on physical and endocrine parameters in stressed mice. It is important to mention that at termination, DMTS may no longer maintain plasma levels as high as at the time of behavioural testing, which may explain the absence of an impact on the parameters.

### 4.3. Lack of TRPA1 Abolishes the Inhibitory Effect of DMTS on CUMS-Induced Depression-like Behaviour of Mice

Gene-knockout mice showed basally elevated anxiety levels in the MBT but not in the OFT. KO animals presented increased depression-like behaviour in the SPT, FST, and TST. Basal anxiety levels were not different between the two genotypes, according to OFT data. Chronic stress caused a similar increase in anxiety levels in KO animals and their WT counterparts. The absence of any basal genotype-related difference in the OFT test results supports the idea that the effects could be specific to the test [48].

Under chronic stress, increased anxiety levels, anhedonia, and reduced activity were detected in wild-type mice indicating depression-like behaviour. The basal depression-like phenotype observed in KO mice might be explained by increased UCN1 and CRH2 receptor signalling in the DR [49,50,51].

The two genotypes differed in their response to stress. KO animals appeared to be unaffected by stress, probably because they already exhibited depression-like behaviour.

CUMS induced depression- and anxiety-like phenomena in the FST, TST, SPT, MBT, and OFT in the naïve stressed group compared to the control group in WT animals. Stress-induced anxiety was prevented by the lack of *Trpa1*.

*Trpa1* KO animals showed no effect of CUMS on depression-like behaviour both in the FST and TST. However, anhedonia developed upon chronic stress in KO animals in the SPT, which is a key symptom of depression [52]. Our results align with those of Kormos et al. [2]. The lack of TRPA1 also prevented the stress-related increase in anxiety levels in the MBT but not in the OFT. The changes observed may be due to increased basal depression-like behaviour in *Trpa1* KO mice, which prevented further increases in depressive parameters.

Conflicting findings were obtained regarding the impact of DMTS on anxiety levels. DMTS treatment can increase baseline anxiety levels of mice via TRPA1 ion channel activation in the OFT. DMTS treatment reduced the anxiety levels of stressed KO animals. The treatment increased baseline anxiety levels in the MBT partly via TRPA1, and it seemed to reduce anxiety levels in stressed, anxious animals in a TRPA1-dependent manner as well. The fact that DMTS had a TRPA1-mediated differential effect on anxiety and depression-like behaviour in animals leads to the proposal that anxiety may be differentially regulated from depression. The TRPA1 ion channel also plays a role in this differential regulation [2]. Previous studies revealed an inhibitory action of H_2_S on anxiety-like behaviour mediated by the mitigation of oxidative stress [53] or regulation of neuroplasticity [54].

Our findings regarding the effect of DMTS on depression-like behaviour were more consistent. DMTS-treated WT mice exhibited reduced anhedonia and depression-like behaviour when exposed to chronic stress. On the other hand, there was no effect of DMTS on depression-like behaviour in KO animals. The treatment did not change their baseline depression-like behaviour. DMTS did not prevent the development of anhedonia in KO mice. The protective effect of DMTS treatment against anhedonia is mediated through the TRPA1 ion channel. Anhedonia extends beyond major depression into the field of anxiety disorders. Anhedonia may be present in generalised anxiety disorder [55]. Our results obtained in the OFT, a test assessing anxiety, as well as in the relative mass of the adrenal glands in KO animals, corroborate these findings. This is important because extant psychological and pharmacological therapies are relatively ineffective against anhedonia. There is an unmet therapeutic need for this high-risk symptom. While traditional psychological and pharmacological interventions for anxiety and depression have primarily targeted the reduction in distress and negative emotions, contemporary models increasingly advocate for a dual approach—simultaneously aiming to alleviate negative symptoms and enhance positive emotional functioning [52].

Overall, the CUMS paradigm resulted in a anxiety and depression-like phenotype, and the response to chronic stress seems to be TRPA1-dependent. DMTS treatment alleviates depression-like behaviour and anxiety. In most cases, this is a TRPA1-independent process. Except for the sucrose preference, TRPA1 is required for DMTS to mitigate anhedonia. One possible underlying mechanism for this is that H_2_S has a TRPA1-independent antagonistic effect on depression-like behaviour mediated by the facilitation of hippocampal long-term potentiation and augmentation of synaptic neurotransmission involved in the regulation of synaptic plasticity [28]. In addition to cognitive impairment, several studies have linked depression and anxiety to alterations in synaptic plasticity [56,57,58].

### 4.4. Effect of DMTS Treatment on Neuronal Activation Depends on Stress Exposure and the Presence of Functional Trpa1

ΔFOSB accumulates after repeated stimulation. Growing evidence suggests that the accumulation is a homeostatic response to chronic stress. The overexpression of ΔFOSB in the brain has been found to enhance resilience to stress, while the inhibition of its activity has been associated with increased susceptibility to stress [59]. The FOSB antibody used in the present study recognises all splice variants of the full-length FOSB and ΔFOSB proteins. The latter isoform is involved in chronic stress adaptation [60,61,62,63].

In our previous study, we performed RNAscope ISH to detect *Trpa1* expression in the lateral septum (LS) and paraventricular nucleus of the thalamus (PVT). RNAscope is more appropriate than immunohistochemistry due to the lack of sufficiently specific commercially available antibodies [64,65,66]. Our findings suggest that the LS and PVT areas do not express *Trpa1* [22].

The basal neuronal activation of the two genotypes was not different in the EWcp, DR, PAG, PVN, and BLA brain areas. However, our findings revealed a fundamental difference between *Trpa1* WT and KO mice. FOSB immunoreactivity in KO non-stressed mice was higher in the PVT but lower in the LS and BST areas compared to their WT counterparts.

The results of our study indicate that exposure to chronic stress significantly increased neuronal activation in the tested brain areas in *Trpa1* WT animals except the BLA. Interestingly, the impact of chronic stress on FOSB expression was observed in the PVN brain area of KO mice, although to a lesser extent than in WT animals. This suggests that the elevated neuronal activation response to chronic stress in PVN is TRPA1-dependent. TRPA1-dependent activation can occur directly or indirectly, depending on whether the ion channel is expressed in the brain area.

In the EWcp, DR, and PAG, we obtained similar results for FOSB immunoreactivity in response to chronic stress regarding the differences observed between genotypes. DMTS treatment reduced the elevated neuronal activity after chronic stress exposure, which is in line with the findings from behavioural tests of depression-like behaviour. Neuronal activation in these brain areas was not significantly higher after chronic stress in DMTS-treated animals. Since no increase in neuronal activation after chronic stress was observed in KO mice, the effect of DMTS cannot be regarded as TRPA1-dependent. The neurons that produce 5-HT in the DR and the cells that produce UCN1 in the EWcp both play a role in the development of mood disorders and anxiety related to stress [67]. When the neurons in the DR are activated, anxiety-like behaviour arises [68]. TRPA1 ion channels were identified in specific cell populations within the DR but not in serotonergic neurons (Milicic et al., in press), implying that DMTS may exert its effects through distinct mechanisms in this region. Urocortin-expressing neurons of the EWcp are functionally connected with serotonergic neurons of the DR, establishing a bidirectional communication between the two nuclei. The DR has CRH receptors. The projection is affected by UCN1 and modulates 5-HT release, affecting mood and anxiety [69,70]. The urocortinergic neurons from the EWcp nucleus project primarily to the DR and PVN areas, which are involved in the stress response [69,71,72]. It is thought that the projections from the DR cause the suppression of PAG cells. Changes in the levels of extracellular 5-HT in the DR have been shown to decrease the panic responses triggered by PAG stimulation [73].

Although the PVT does not express the ion channel [22], we noticed a difference in basal neuronal activation between the two genotypes, which did not change in response to chronic stress. Notably, a subset of efferent projections from TRPA1-expressing neurons in the EWcp innervates the PVT. This neuronal input may explain the difference in basal neuronal FOSB immunoreactivity in KO animals [74]. The PVT is heavily influenced by input from the hypothalamus, which includes a dense network of neuropeptide-containing neurons [75]. Prefrontal cortical areas, such as the infralimbic, prelimbic, and insular cortices, play a crucial role in providing input to the PVT. The dorsomedial nucleus of the hypothalamus, the PAG, and the lateral parabrachial nucleus also contribute to the input received by the PVT. Additionally, the PVT receives projections from various neurons located throughout the brainstem and hypothalamus. These widespread projections to the PVT demonstrate remarkable diversity in neurochemical contents, including fibrous projections involving monoamines (such as dopamine, noradrenaline, adrenaline, and serotonin (5-HT)), as well as an extensive array of neuropeptides [76]. The neuronal-activity-enhancing effect of chronic stress was observed in the PVT, as well. Neuronal activation did not increase further in response to chronic stress in KO mice, maybe because it was already elevated. The effect of DMTS was not statistically significant in the PVT area. There was a tendency for DMTS to reduce neuronal activation in wild-type mice exposed to chronic stress.

The LS is recognised as a critical structure involved in the regulation of emotional processing and stress responsiveness. Some studies suggest that the LS promotes coping behaviour during active stress and is involved in an HPA axis inhibitory mechanism mediated, at least in part, by septal 5-HT1A receptors and does not undergo glucocorticoid-mediated feedback [77]. Even though *Trpa1* is not present in the LS area [22], we still noted a difference in the baseline neuronal activation of *Trpa1* KO and WT mice in that brain region following chronic stress and DMTS treatment. TRPA1 may be involved in facilitating the activation-boosting effect of DMTS through an indirect mechanism. The LS is reached by a highly intense urocortinergic fibre network from the EWcp [78]. It is known that UCN1 cells express the highest amount of *Trpa1* in the mouse brain [2]. The absence of TRPA1 in the EWcp nucleus in KO animals may explain the findings in the LS region. DMTS treatment did not decrease neuronal activation in stressed KO animals, and chronic stress did not increase it.

No basal difference in neuronal activation was observed between the two genotypes in the PVN area. Chronic stress strongly increased neuronal activation in WT and KO animals despite the PVN expressing *Trpa1* (Milicic et al., under publication). Other processes are probably involved in adapting the PVN to chronic stress besides TRPA1. The PVN is a region that is essential for the initiation of stress responses. Chronic stress induces changes in the expression of peptides, neurotransmitter receptors, and neuronal excitability in the PVN [79,80,81]. PVN neurons are hyperreactive and hyperresponsive to stimuli following repeated stress, contributing to the acute hormone-release-promoting effect of chronic stress. Chronic stress increases the number of presynaptic excitatory neurotransmitter boutons (glutamate and norepinephrine) targeting CRH cell bodies and dendrites [82]. We noticed a reduction in neuronal activity in WT animals following the application of DMTS, indicating that the TRPA1 ion channel is at least partly involved in mediating the effects of DMTS in the PVN region. It is important to note that changes in the SPT after DMTS application and chronic stress exposure are consistent with changes in the PVN. Considering that the PVN expresses the TRPA1 ion channel and the SPT test showed no effect of DMTS in the KO animals, the positive action of DMTS on anhedonia levels may be due to the activation of TRPA1 ion channels.

There was a significant difference in neuronal activation between the two genotypes in the BST region, with lower activation in the gene knockout mice. Neuronal activation was increased in response to chronic stress. Similarly to the LS and PVT areas, we found a difference in basal and post-chronic-stress neuronal activity in the BST area between WT and KO animals. DMTS treatment had no effect on neuronal activation in this brain region in either group. The BST is believed to be a hub connecting limbic cognitive centres and nuclei involved in processing reward, stress, and anxiety. It also receives incoming information about systemic stressors like hypertension and haemorrhage [83]. As a result, the BST appears to have an important role in coordinating physiological and behavioural responses, linking structures such as the amygdala, hippocampus, and medial prefrontal cortex (MPFC) to hypothalamic and brainstem regions involved in autonomic and neuroendocrine functions [84]. Some peptides present in the BST, such as pituitary adenylate cyclase-activating polypeptide (PACAP), can directly activate CRH neurons. There may be potential excitatory peptidergic projections to the PVN [85]. PACAP is up-regulated in the BST by chronic stress, leading to hyperactivity of the HPA axis [78]. The impact of the BST on how the body responds to stress depends on how long the exposure to stress lasts. For instance, when the ventral part of the BST is damaged, it reduces the response to acute stress, but it significantly increases the stress response of the body after prolonged exposure to unpredictable stress. Different groups of cells may be activated by sudden stress compared to prolonged ones. Damage to the dorsal part of the BST heightens the stress response after prolonged exposure to unpredictable stress, indicating its role in keeping the stress system from becoming too sensitive after prolonged stress [86].

There was no difference in neuronal activation between the two genotypes in the BLA. Stress had no effect in either group, and DMTS treatment did not affect neuronal activation in this brain region. The BLA is a critical brain region involved in fear and anxiety [87]. Recent studies suggest that the medial amygdala (MeA), BLA, and central amygdala (CeA) do not exhibit increased FOSB/ΔFOSB expression in response to acute restraint stress or chronic variable/repeated stress [88].

It is also worth mentioning that the magnitude of neuronal activation was specific to the given brain area. We detected an almost four-fold elevation in FOSB expression within the EWcp and DR, while levels in the PAG and PVN increased by approximately threefold. We observed a 2–2.5-fold increase in the PVT and BST and only a 1.5-fold increase in the LS.

Summarising the effects of DMTS treatment, we observed that it did not impact FOSB expression in KO mice. However, FOSB immunoreactivity was significantly lower in WT DMTS-treated animals than in their stressed but untreated counterparts in all brain areas that we examined, except in the BST and BLA. Notably, the magnitude of the impact of DMTS treatment was also brain-area-specific. Brain areas known to express TRPA1 (EWcp, PAG, DR) exhibited the largest effect upon DMTS administration. Brain areas that do not express the TRPA1 (LS, PVT) ion channel also responded to DMTS, but the effect was 2–3x less. In brain areas that do not express *Trpa1*, indirect activation from an area expressing TRPA1 may be the explanation [74]. TRPA1 is only partially involved in mediating the effects of DMTS in stress adaptation among other processes.

The protective effect of sulphides, specifically that of H_2_S, in the CNS has been previously described. They have been studied in various neurodegenerative diseases, focusing on protein S-persulfidation [89], the role of sulphides in neuroprotection through the TRPA1 ion channel [90], or its participation in neuromodulation through the reduction of endoplasmic reticulum (ER) stress [91]. The H_2_S used in previous studies cannot directly modify cysteines via persulfidation. Polysulfides, such as alkyl trisulfides like DMTS, can directly participate in the reaction [92]. Chemically, alkyl trisulfides form tri- and disulfide metabolites with thiol groups of cysteine amino acids, while inorganic polysulfides lead to the persulfidation of proteins. For this reason, the two groups of compounds (organic and inorganic polysulfides) may have different biological effects. DMTS has an elimination half-life of 36 min and accumulates mainly in red blood cells. It readily crosses the blood–brain barrier and can be detected in brain tissue. It is also patented in the United States of America as an intramuscular antidote for cyanide poisoning [8,24,93,94,95,96].

The changes in physical/endocrine, behavioural, and neuronal activation parameters in response to chronic stress and DMTS treatment can be related in some instances. In *Trpa1* wild-type mice, anxiety and depression-like behaviour under chronic stress were measured. DMTS attenuated all of these effects.

We observed correlations between relative adrenal weights, SPT findings, OFT results and PVN brain area activation changes after chronic stress in gene knockout mice. The relative adrenal weight, depression, and anxiety parameters in gene-knockout animals were not affected by DMTS. The chronic-stress-induced anhedonia assessed in the SPT is not a TRPA1-dependent process, but TRPA1 must partially mediate the reduction in anhedonia after DMTS treatment. In addition, based on the neuronal activation pattern in the PVN, this brain area mediates the antidepressant effect of DMTS on anhedonia in a TRPA1-dependent manner.

The antidepressant and anxiolytic effects of DMTS may be mediated through modulation of the HPA axis and may be mediated by the EWcp, DR, PAG, LS, and PVN either directly or indirectly through the TRPA1 ion channel.

Based on current evidence, DMTS appears to selectively modulate TRPA1 among the TRP channel family. This selectivity likely stems from the unique molecular architecture of TRPA1, which contains multiple reactive cysteine residues in its N-terminal domain. These thiol groups serve as key sites for covalent modification by electrophilic agents such as DMTS, which can form disulfide or polysulfide bonds and thereby induce channel activation. In contrast, most other TRP channels either lack comparable cysteine-rich motifs or are structurally resistant to electrophilic modulation, and no compelling data support their activation by DMTS [7]. While TRPV1 is expressed in several CNS regions and could be considered a potential target, our previous work demonstrated that DMTS does not elicit calcium influx in TRPV1-positive CHO cells, supporting functional selectivity for TRPA1 [18]. Furthermore, although TRPV3 has been shown to respond to certain electrophilic stimuli, its expression is predominantly restricted to peripheral tissues such as keratinocytes, with a negligible presence in the central nervous system [97]. Taken together, these data strongly support a TRPA1-specific mechanism of action for DMTS within the CNS.

It is important to clarify that vehicle-treated and naïve control groups did not differ significantly in any behavioural or in vitro outcomes, indicating that repeated i.p. injections in the absence of DMTS did not influence stress-related parameters. This observation is consistent with previous findings showing that injection-induced acute stress typically resolves within 24 h [98,99].

A brief summary of our main findings (Figure 13):*Trpa1* deficiency affects/modifies the response to chronic stress, both at the level of behaviour and neuronal activation.CUMS leads to anxiety and depression-like behaviour, dysregulation of the HPA axis, and activation of brain areas important in stress adaptation in a TRPA1-dependent manner.DMTS alleviates depression-like behaviour in *Trpa1* WT mice.The lack of TRPA1 reverses the antidepressant-like action of DMTS in chronic stress-exposed mice according to the SPT test.DMTS alleviates thymus involution caused by dysregulation of the HPA axis due to chronic stress.The action of DMTS on anxiety and depression-like behaviour may be differentially and independently regulated.The absence of the TRPA1 ion channel mitigates the inhibitory effect of DMTS on chronic-stress-induced neuronal activation in PVN.

**Figure 13 pharmaceutics-17-00781-f013:**
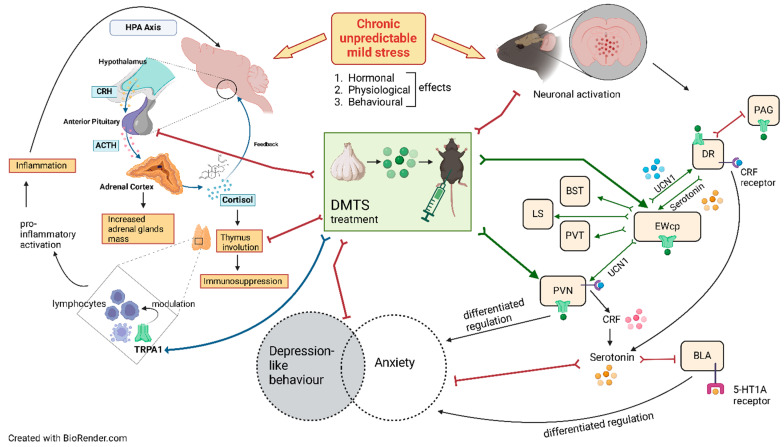
Summary of possible mechanisms underlying the effects of chronic stress and DMTS treatment on depression-like behaviour and anxiety. The antidepressant-like effects of DMTS (e.g., in the sucrose preference test) were abolished in *Trpa1*-knockout mice, suggesting a TRPA1-dependent mechanism, whereas its anxiolytic-like effects (e.g., increased peripheral zone activity in the open field test) were preserved, indicating TRPA1-independent pathways, possibly involving serotonergic circuits in limbic regions, such as the basolateral amygdala. Red arrows = inhibition; green arrows = stimulation; blue arrow = modulation; black arrow = potential regulation. HPA = hypothalamic–pituitary–adrenal axis; TRPA1 = transient receptor potential ankyrin 1; ACTH = adrenocorticotropic hormone; CRH = corticotropin-releasing hormone; DMTS = dimethyl trisulfide; CRF = corticotropin-releasing factor; 5-HT1A = serotonin 1A; UCN1 = urocortin 1; PAG = periaqueductal grey matter; DR = dorsal raphe nucleus; EWcp = centrally projecting Edinger–Westphal nucleus; BST = bed nucleus of the stria terminalis; BLA = basolateral amygdala; LS = lateral septum; PVT = paraventricular nucleus of the thalamus; PVN = paraventricular nucleus of the hypothalamus.

## 5. Conclusions

We confirmed the role of the TRPA1 ion channel in stress-adaptation processes using a CUMS mouse model. We identified the potential anti-anxiety and anti-depressant effects of the naturally occurring organic polysulfide DMTS and the partial role of the TRPA1 ion channel in mediating these effects. We identified the brain areas mediating the effects of DMTS on stress adaptation processes and the extent of the impact in each brain area.

Our findings provide novel insight into the mechanism of the ameliorating effect of DMTS on depression and anxiety and suggest that TRPA1 is a promising therapeutic target for mood disorders, which exploits new paths for the prevention and cure of depression under chronic stress.

Based on all this, DMTS is an ideal candidate for further investigation as a dietary supplement or complementary therapeutic in the treatment of anxiety and depression.

## 6. Limitations

It is necessary to point out the shortcomings of our experiments. The behavioural profile noted in our global KO mouse line may reflect compensatory adaptations arising during development. It remains possible that additional, unexamined peripheral or central pathways contributed to the phenotype, given that the functional receptor was ablated in both systemic and central compartments [2,100,101]. Female subjects and corresponding tissue samples were excluded from the present investigation. The oestrous cycle may influence the responses in female mice, as some of the brain areas that we examined express the oestrogen receptor ß [102,103,104,105,106,107,108]. Another limitation is related to the substance used in our research study. DMTS has not been investigated in behavioural tests previously, except in our recently published paper [22], and consequently, available data on the potential effects of DMTS on the central nervous system and behavioural outcomes remain limited.

## 7. Future Plans

Building on our current findings, future research will explore the neurobiological mechanisms underlying the anxiolytic and antidepressant-like effects of DMTS, with a specific focus on the endocannabinoid system. In parallel, molecular and immunohistochemical analyses will assess region-specific expression changes in stress-related brain areas. Additionally, we plan to investigate the systemic and organ-specific toxicity profile of DMTS and related polysulfides. In addition, future studies will assess the efficacy and pharmacokinetics of transdermal DMTS application, as a non-invasive delivery route with potential translational relevance. These complementary approaches will support both mechanistic insight and translational evaluations for potential therapeutic use.

## Figures and Tables

**Figure 1 pharmaceutics-17-00781-f001:**
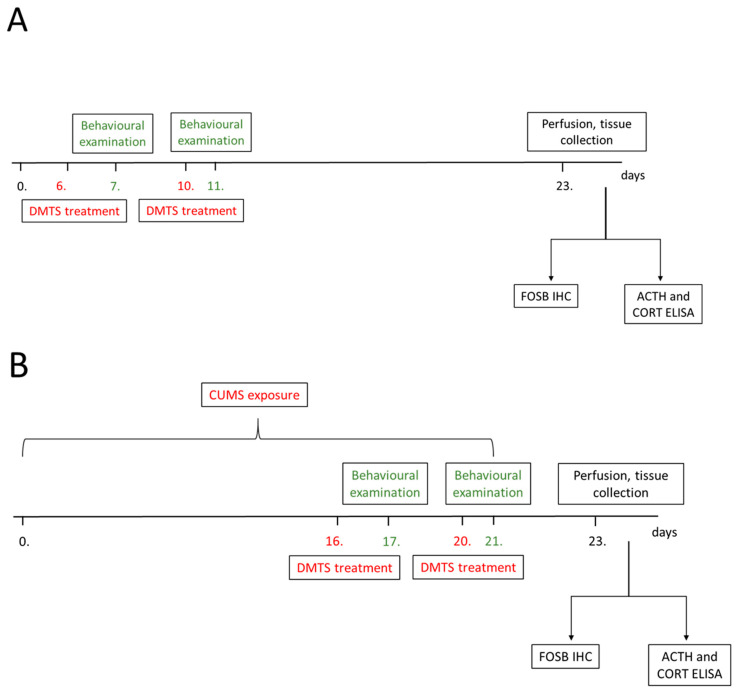
Experimental timeline of control (**A**) and chronic-stress-exposed (**B**) animals. IHC: immunohistochemistry; ACTH: adrenocorticotropic hormone; CORT: corticosterone; CUMS: chronic unpredictable mild stress; DMTS: dimethyl trisulfide; ELISA: enzyme-linked immunosorbent assay.

**Figure 2 pharmaceutics-17-00781-f002:**
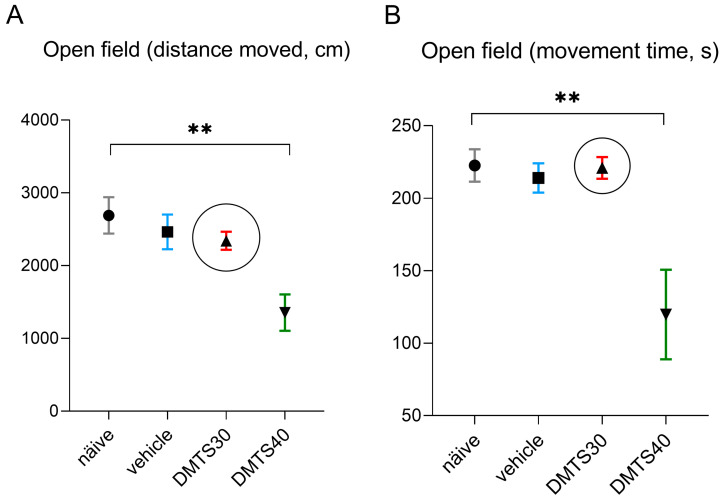
Establishing the dose of dimethyl trisulfide (DMTS) via open field tests. Symbols represent, from left to right, untreated (naïve), vehicle-treated, 30 mg/kg (DMTS30), and 40 mg/kg DMTS-treated (DMTS40) *Trpa1* WT mice. The distance travelled (**A**) and mobility time (**B**) are plotted on the left y-axes (one-way analysis of variance; ** *p* < 0.01; *n* = 8–10 per group).

**Figure 3 pharmaceutics-17-00781-f003:**
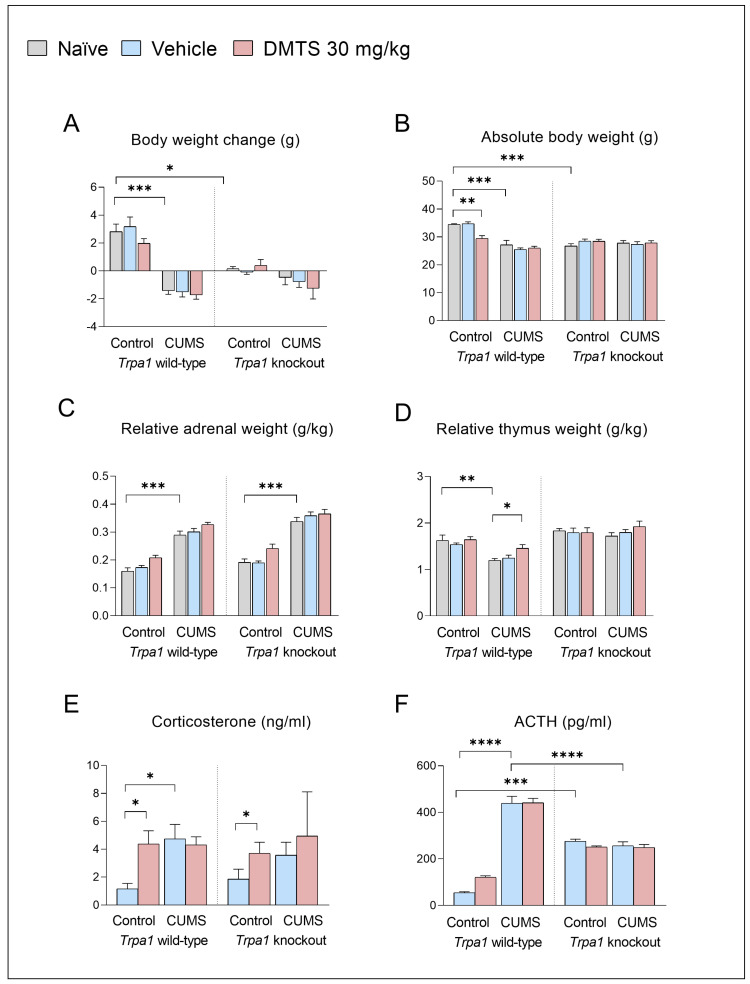
The efficacy of chronic unpredictable mild stress (CUMS) exposure and dimethyl trisulfide (DMTS) treatment. We found that the body weight change (**A**), relative adrenal weight (**C**), and relative thymus weight (**D**) mirrored the somatic changes induced by chronic stress. Panel (**B**) presents absolute body weights of mice at the end of the in vivo experiment. We assessed the hypothalamic–pituitary–adrenal axis activity by determining the serum corticosterone (**E**) and adrenocorticotropic hormone (ACTH) concentrations (**F**) (three-way analysis of variance followed by Tukey’s post hoc test; * *p* < 0.05; ** *p* < 0.01; *** *p* < 0.001; **** *p* < 0.0001; *n* = 8–16 per group). Only the relevant significance levels have been indicated to facilitate clarity.

**Figure 4 pharmaceutics-17-00781-f004:**
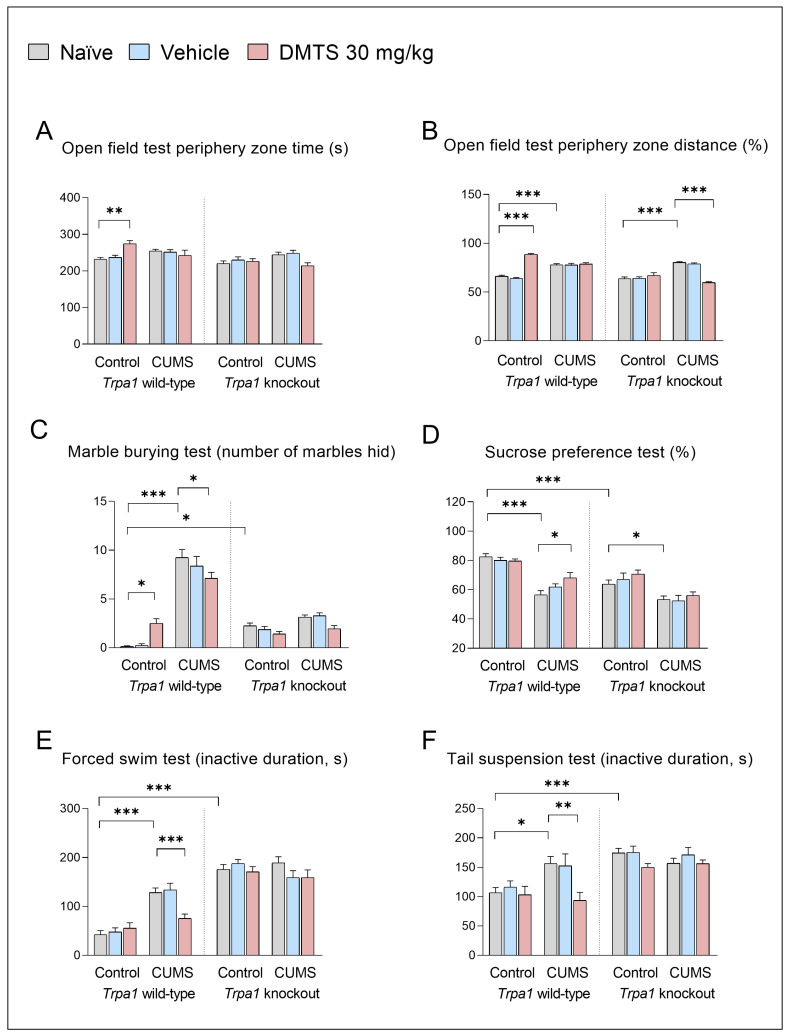
Effect of chronic unpredictable mild stress (CUMS) and dimethyl trisulfide (DMTS) treatment on anxiety and depression-related behaviour. In order to assess the level of anxiety, we evaluated the time spent (**A**) and the distance travelled (**B**) in the periphery part of the arena in the open field test, and we counted the number of hidden marbles in the buried marble test (**C**). To detect depression-like behaviour, we determined the presence of anhedonia with the sucrose preference test (**D**); moreover, we assessed the inactive duration in the forced swim test (**E**) and tail suspension test (**F**) (three-way analysis of variance followed by Tukey’s post hoc test; * *p* < 0.05; ** *p* < 0.01; *** *p* < 0.001; *n* = 8–16 per group). Only the relevant significance levels have been indicated to facilitate clarity.

**Figure 5 pharmaceutics-17-00781-f005:**
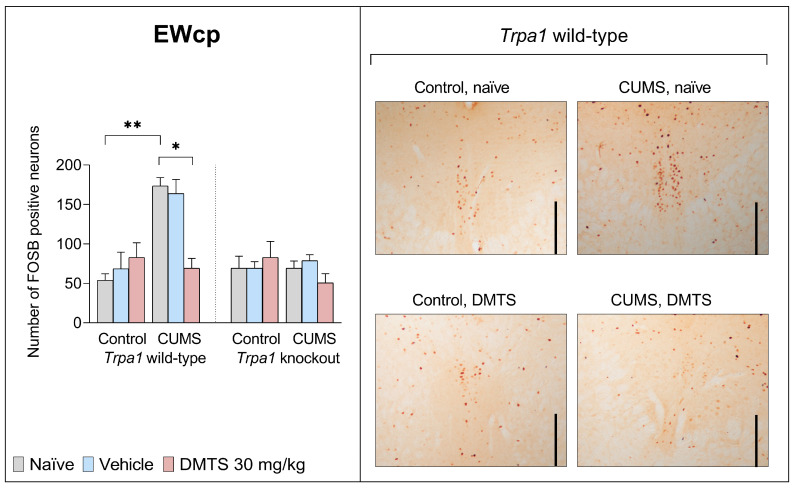
FOSB immunohistochemistry in centrally projecting Edinger–Westphal nucleus (EWcp), representative micrographs. Bar graphs show the number of FOSB-immunoreactive neurons in each group. Three-way analysis of variance followed by Tukey’s post hoc test: * *p* < 0.05; ** *p* < 0.01; *n* = 5–6 per group. Black line: scale bar 200 μm. Control = non-stressed; CUMS = chronic unpredictable mild stress; DMTS = dimethyl trisulfide. Only the relevant significance levels have been indicated to facilitate clarity.

**Figure 6 pharmaceutics-17-00781-f006:**
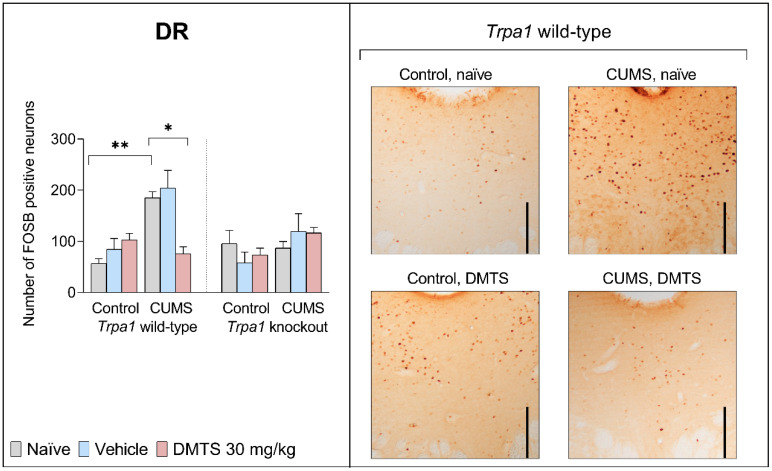
FOSB immunohistochemistry in dorsal raphe nucleus (DR), representative micrographs. Bar graphs show the number of FOSB-immunoreactive neurons in each group. Three-way analysis of variance followed by Tukey’s post hoc test: * *p* < 0.05; ** *p* < 0.01; *n* = 5–6 per group. Black line: scale bar 200 μm. Control = non-stressed; CUMS = chronic unpredictable mild stress; DMTS = dimethyl trisulfide. Only the relevant significance levels have been indicated to facilitate clarity.

**Figure 7 pharmaceutics-17-00781-f007:**
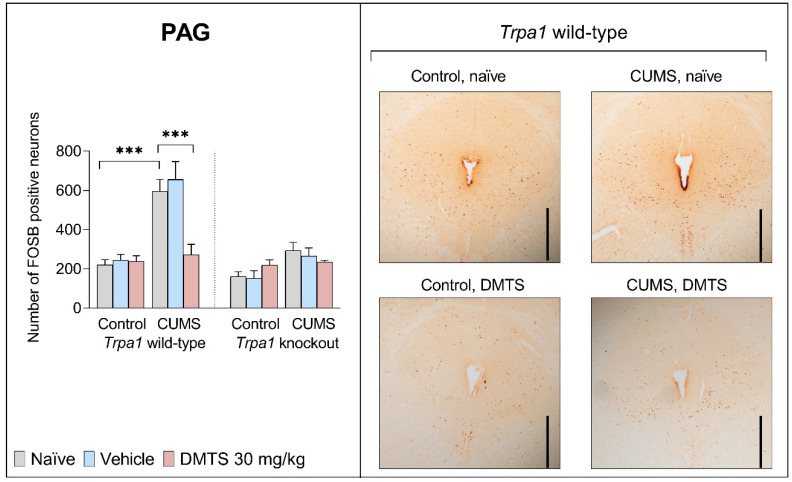
FOSB immunohistochemistry in periaqueductal grey matter (PAG), representative micrographs. Bar graphs show the number of FOSB-immunoreactive neurons in each group. Three-way analysis of variance followed by Tukey’s post hoc test: *** *p* < 0.001; *n* = 5–6 per group. Black line: scale bar 500 μm. Control = non-stressed; CUMS = chronic unpredictable mild stress; DMTS = dimethyl trisulfide. Only the relevant significance levels have been indicated to facilitate clarity.

**Figure 8 pharmaceutics-17-00781-f008:**
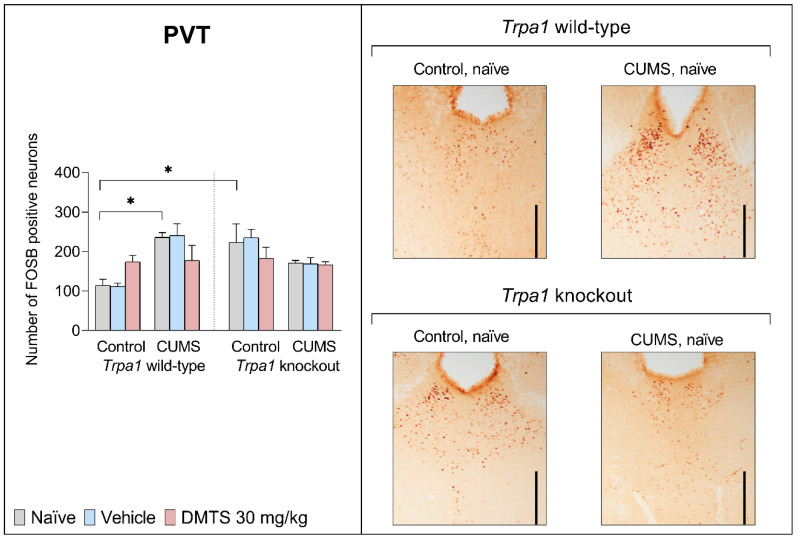
FOSB immunohistochemistry in paraventricular nucleus of the thalamus (PVT), representative micrographs. Bar graphs show the number of FOSB-immunoreactive neurons in each group. Three-way analysis of variance followed by Tukey’s post hoc test: * *p* < 0.05; *n* = 5–6 per group. Black line: scale bar 200 μm. Control = non-stressed; CUMS = chronic unpredictable mild stress; DMTS = dimethyl trisulfide. Only the relevant significance levels have been indicated to facilitate clarity.

**Figure 9 pharmaceutics-17-00781-f009:**
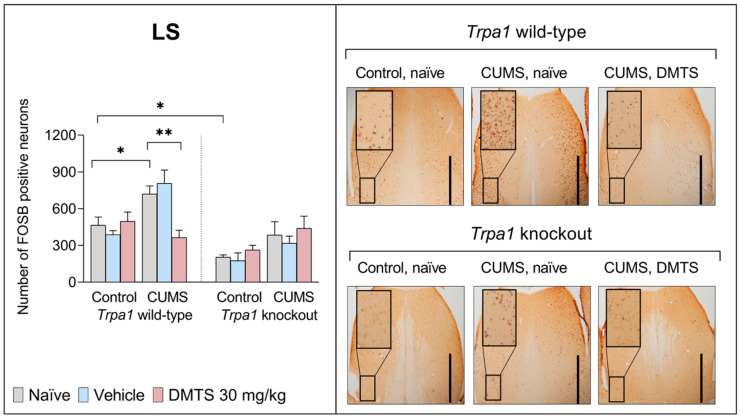
FOSB immunohistochemistry in lateral septal nucleus (LS), representative micrographs. Bar graphs show the number of FOSB-immunoreactive neurons in each group. Three-way analysis of variance followed by Tukey’s post hoc test: * *p* < 0.05, ** *p* < 0.01; *n* = 5–6 per group. Black line: scale bar 500 μm. Control = non-stressed; CUMS = chronic unpredictable mild stress; DMTS = dimethyl trisulfide. Only the relevant significance levels have been indicated to facilitate clarity.

**Figure 10 pharmaceutics-17-00781-f010:**
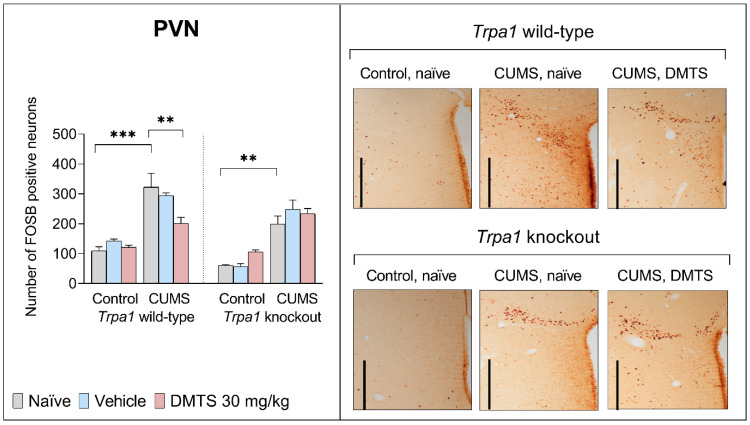
FOSB immunohistochemistry in paraventricular nucleus of the hypothalamus (PVN), representative micrographs. Bar graphs show the number of FOSB-immunoreactive neurons in each group. Three-way analysis of variance followed by Tukey’s post hoc test: ** *p* < 0.01, *** *p* < 0.001; *n* = 5–6 per group. Black line: scale bar 200 μm. Control = non-stressed; CUMS = chronic unpredictable mild stress; DMTS = dimethyl trisulfide. Only the relevant significance levels have been indicated to facilitate clarity.

**Figure 11 pharmaceutics-17-00781-f011:**
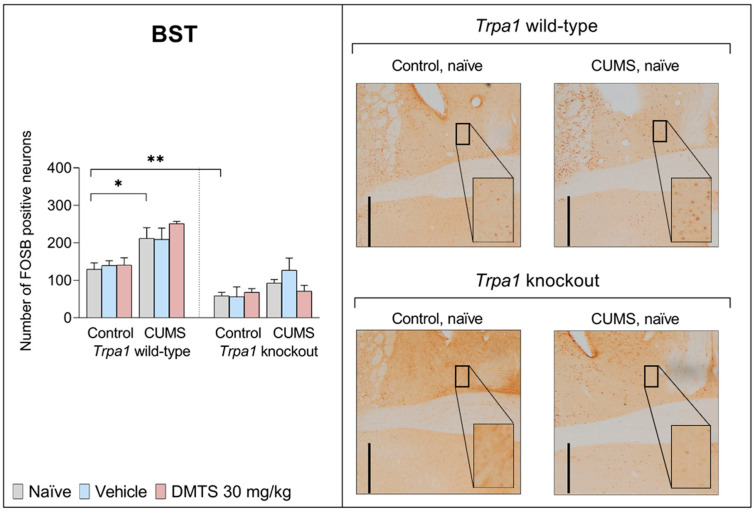
FOSB immunohistochemistry in bed nucleus of the stria terminalis (BST), representative micrographs. Bar graphs show the number of FOSB-immunoreactive neurons in each group. Three-way analysis of variance followed by Tukey’s post hoc test: * *p* < 0.05, ** *p* < 0.01; *n* = 5–6 per group. Black line: scale bar 500 μm. Control = non-stressed; CUMS = chronic unpredictable mild stress; DMTS = dimethyl trisulfide. Only the relevant significance levels have been indicated to facilitate clarity.

**Figure 12 pharmaceutics-17-00781-f012:**
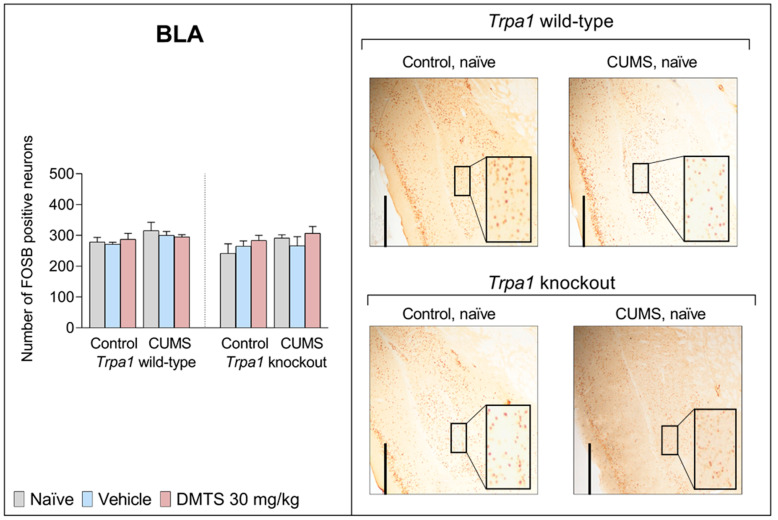
FOSB immunohistochemistry in basolateral amygdala (BLA), representative micrographs. Bar graphs show the number of FOSB-immunoreactive neurons in each group. Three-way analysis of variance followed by Tukey’s post hoc test: *n* = 5–6 per group. Black line: scale bar 500 μm. Control = non-stressed; CUMS = chronic unpredictable mild stress; DMTS = dimethyl trisulfide. Only the relevant significance levels have been indicated to facilitate clarity.

**Table 1 pharmaceutics-17-00781-t001:** The table summarises the treatment groups and the element numbers. *Trpa1*: transient receptor potential ankyrin 1; WT: wild-type; KO: knockout; CUMS: chronic unpredictable mild stress; DMTS30: dimethyl trisulfide 30 mg/kg dose; naïve: non-treated.

** *Trpa1* ** **WT**					
Non-stressed			CUMS		
naïve	vehicle-treated	DMTS30-treated	naïve	vehicle-treated	DMTS30-treated
8	16	16	8	16	16
** *Trpa1* ** **KO**					
Non-stressed			CUMS		
naïve	vehicle-treated	DMTS30-treated	naïve	vehicle-treated	DMTS30-treated
8	16	16	8	16	16

**Table 2 pharmaceutics-17-00781-t002:** Timeline of chronic unpredictable mild stress protocol with mid-day and overnight stressors. DARK = dark room; GH = group holding; REST = restraint stress; SHAKE = shaker stress; ISOL = social isolation; TILT = tilted cage; WM: body weight measurement; WET = wet bedding.

No. of Days	Mid-Day Stressors	Overnight Stressors
1.	WM, REST	WET
2.	DARK	ISOL
3.	TILT	GH
4.	DARK	WET
5.	SHAKE	GH
6.	TILT	ISOL
7.	REST	GH
8.	TS, TILT	WET
9.	SHAKE	ISOL
10.	REST	WET
11.	TILT	GH
12.	SHAKE	WET
13.	DARK	GH
14.	TILT	WET
15.	WM, REST	GH
16.	TILT	WET
17.	REST	GH
18.	DARK	WET
19.	TILT	GH
20.	DARK	WET
21.	REST	ISOL

**Table 3 pharmaceutics-17-00781-t003:** Summary of the statistics of physical parameters and endocrine and behavioural data based on three-way analysis of variance (ANOVA). ACTH = adrenocorticotropic hormone; CUMS = chronic unpredictable mild stress. Significant values are highlighted in bold.

		Main Effects			Interactions			
Variable		Treatment	Genotype	CUMS	Treatment × Genotype	CUMS × Treatment	CUMS × Genotype	CUMS × Treatment × Genotype
**MBT, number of marbles hid**	F_2,149_	0.9901	**57.66**	**186.3**	2.325	**8.423**	**110.6**	**5.556**
	*p*	0.374	**<0.0001**	**<0.0001**	0.1013	**0.0003**	**<0.0001**	**0.0047**
**OFT, periphery zone time**	F_2,72_	0.1927	**13.57**	1.654	**4.723**	**45583**	0.7645	0.3862
	*p*	0.8252	**0.0004**	0.204	**0.0118**	**0.0002**	0.3859	0.6815
**OFT, periphery zone distance**	F_2,72_	**3.657**	**30498**	**46.64**	**93.32**	**61.24**	2.189	0.3012
	*p*	**0.0333**	**<0.0001**	**<0.0001**	**<0.0001**	**<0.0001**	0.1455	0.7413
**FST, activity**	F_2,89_	**3.923**	**148.1**	**18.73**	0.7528	1.389	**53.56**	**5.322**
	*p*	**0.0233**	**<0.0001**	**<0.0001**	0.474	0.2559	**<0.0001**	**0.007**
**TST, activity**	F_2,73_	**5.814**	**45322**	1.963	0.9288	0.8436	**4.864**	**4.135**
	*p*	**0.0045**	**<0.0001**	0.1678	0.3997	0.4366	**0.0323**	**0.0222**
**SPT, sucrose preference**	F_2,78_	**3.172**	**13516**	**31079**	0.02048	45304	2.555	**3.496**
	*p*	**0.0474**	**<0.0001**	**<0.0001**	0.9797	0.3292	0.1146	**0.0359**
**Serum ACTH**	F_1,16_	0.7709	0.3015	**243.3**	**5.494**	1.222	**275.1**	3.534
	*p*	0.3929	0.5905	**<0.0001**	**0.0323**	0.2854	**<0.0001**	0.0785
**Serum corticosterone**	F_1,30_	**4.197**	0.03008	**4.84**	0.02317	1.975	0.03587	1.174
	*p*	**0.0493**	0.8635	**0.0357**	0.88	0.1702	0.8511	0.2872
**Relative adrenal weight**	F_1,56_	**10.87**	**25.35**	**415.3**	0.009887	1.105	2.79	0.5117
	*p*	**<0.0001**	**<0.0001**	**<0.0001**	0.9902	0.3383	0.1004	0.6023
**Relative thymus weight**	F_2,121_	2.248	**55.79**	**9.346**	0.3015	2.179	**10.35**	0.01058
	*p*	0.11	**<0.0001**	**0.0028**	0.7402	0.1176	**0.0017**	0.9895
**Change in body weight**	F_2,107_	15.02	**14.43**	**45.22**	2.162	0.969	**13.44**	0.7456
	*p*	0.1215	**0.0002**	**<0.0001**	0.1201	0.3833	**0.0004**	0.4773
**Body weight**	F_2,77_	2.326	**13.49**	**50.72**	**5.797**	**3.747**	**43.66**	**3.461**
	*p*	0.1045	**0.0004**	**<0.0001**	**0.0045**	**0.028**	**<0.0001**	**0.0364**

**Table 4 pharmaceutics-17-00781-t004:** Summary of statistical analyses of FOSB immunoreactivity in various stress-related brain areas. The analysis was performed via three-way analysis of variance (ANOVA). EWcp = centrally projecting Edinger–Westphal nucleus; LS = lateral septal nucleus; PAG = periaqueductal grey matter; DR = dorsal raphe nucleus; PVN = paraventricular nucleus of the hypothalamus; PVT = paraventricular nucleus of the thalamus; BLA = basolateral amygdala; BST = bed nucleus of the stria terminalis; CUMS = chronic unpredictable mild stress. Significant values are highlighted in bold.

		Main Effects			Interactions			
Variable		Treatment	Genotype	CUMS	Treatment × Genotype	CUMS × Treatment	CUMS × Genotype	CUMS × Treatment × Genotype
**EWcp**	F_2,27_	2.708	**14.12**	**12.71**	11.09	**1.611**	**22.19**	**3.491**
	*p*	0.0848	**0.0008**	**0.0014**	0.0003	**0.2182**	**<0.0001**	**0.0448**
**LS**	F_2,27_	0.5911	**16.68**	36.4	**3.381**	**4.289**	0.02364	4.091
	*p*	0.5616	**0.0004**	<0.0001	**0.0509**	**0.0255**	0.8791	0.0296
**PAG**	F_2,27_	**4.024**	**58.24**	**29.59**	**10.78**	**5.261**	15.65	45378
	*p*	**0.0311**	**<0.0001**	**<0.0001**	**0.0005**	**0.0128**	0.0006	0.0555
**DR**	F_2,27_	**1.434**	**20.61**	**5.002**	4.278	2.267	**3.268**	**6.678**
	*p*	**0.2559**	**0.0002**	**0.0338**	0.0276	0.123	**0.085**	**0.0057**
**PVN**	F_2,27_	**0.2366**	**166**	16.27	5.995	4.281	**0.03658**	**3.206**
	*p*	**0.7911**	**<0.0001**	0.0005	0.0077	0.0257	**0.8499**	**0.0583**
**PVT**	F_2,27_	**0.4047**	**1.945**	**1.356**	0.8528	0.4163	20.27	**3.702**
	*p*	**0.6717**	**0.1759**	**0.2557**	0.4387	0.6642	0.0001	**0.0397**
**BLA**	F_2,27_	1.146	3.03	**2.553**	**0.7379**	1.137	**0.04659**	0.4684
	*p*	0.3347	0.0945	**0.1232**	**0.4887**	0.3375	**0.8309**	0.6316
**BST**	F_2,27_	**0.2597**	39.09	**62.11**	0.1842	1.029	6.838	2.529
	*p*	**0.7733**	<0.0001	**<0.0001**	0.833	0.3715	0.0158	0.1026

## Data Availability

The datasets used and/or analysed during the current study are available from the corresponding author upon reasonable request.

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
