# Peer review of "Anxiolytic and Antidepressant Effects of Organic Polysulfide, Dimethyl Trisulfide Are Partly Mediated by the Transient Receptor Potential Ankyrin 1 Ion Channel in Mice"

_pharmaceutics, 2025, doi:10.3390/pharmaceutics17060781_

Round 1
Reviewer 1 Report
Comments and Suggestions for Authors
This manuscript, "Anxiolytic and antidepressant effects of organic polysulfide, dimethyl trisulfide are partly mediated by the transient receptor potential ankyrin 1 ion channel in mice," investigates a significant area of neuropsychopharmacology. The study explores the potential of dimethyl trisulfide (DMTS), a naturally occurring organic polysulfide, as an anxiolytic and antidepressant agent.
While the abstract mentions TRPA1 is "known for its role in stress adaptation", the introduction could more clearly synthesize existing knowledge on TRPA1's role in the CNS and specifically in anxiety/depression models before introducing DMTS. This would better set the stage for the hypothesis.
The paper determines the dose based on the Open Field Test (OFT) to avoid motor impairment. It would be beneficial to briefly discuss if this dose has been used in other CNS-related studies with DMTS (if any exist beyond their cited preprint ) or similar polysulfides, to provide further context for its pharmacological relevance.
Fig 13 is a very helpful summary figure. Some arrows indicate stimulation/inhibition/modulation. It would be beneficial if the legend or discussion explicitly linked which parts of the proposed mechanism are considered TRPA1-dependent versus independent, based on their findings. For example, the modulation of "Depression-like behaviour" and "Anxiety" by DMTS could have annotations indicating the degree of TRPA1 involvement found.
Page 1, Abstract, Line 27: "Results showed that CUMS induces depression-like behaviour in a TRPA1-dependent manner" – the data also show KOs have baseline differences, so CUMS effect isn't solely TRPA1-dependent in its entirety but the response to CUMS is altered by TRPA1 status. Perhaps rephrase for precision.
The list of highlights (page 1) is good. Highlight 2 states "Lack of TRPA1 reverses the antidepressant-like action of DMTS." The results suggest it abolishes or prevents the effect rather than reverses it (i.e., making DMTS pro-depressant). Consider rephrasing for accuracy.
The study is deemed significant for its novel investigation into DMTS as a potential treatment for anxiety and depression, and its exploration of the TRPA1 channel's role. The comprehensive methodology, novel findings, and potential clinical relevance are strong points for acceptance. Key suggestions for improvement include enhancing clarity in the introduction regarding TRPA1, providing more rationale for the DMTS dosing regimen, further elaborating on the interpretation of FOSB data and the differential regulation of anxiety versus depression. Expanding the limitations section and refining some statements for precision are also recommended. Overall, the paper makes a valuable contribution to neuropsychopharmacology.
Author Response
Our point-by-point responses to the Reviewer are provided in the attached Word document.

Reviewer 2 Report
Comments and Suggestions for Authors
As for me, this is the systematic research, conducted on the modern level. Conclusions correspond well with the gathered data. I just do not understand the way of citing references - [20] and [21] predecess [2]. What is the reason?
Conclusion - minor corrections
Author Response

(The authors gave the same response as above.)

Reviewer 3 Report
Comments and Suggestions for Authors
This manuscript presents a new investigation into the role of dimethyl trisulfide (DMTS) and the TRPA1 ion channel in modulating depression- and anxiety-like behaviors induced by stress in mice. The study addresses an important gap in understanding the central nervous system (CNS) effects of DMTS. The multi-faceted approach (behavioral, physiological, endocrine, and neuroanatomical) is a strength. However, many revisions are required.
- In the abstract, add missed information about DMTS Administration: Route of administration, dosage, frequency, and treatment duration relative to CUMS . This is very important for reproducibility and efficacy.
- In line 25 in the abstract, FOSB immunohistochemistry, write full name for FOSM.
- How was FOSB signal quantified (e.g., cell counting, integrated density)? is there evidence for this use in the literature?
- In the abstract, Mention duration for CUMS Protocol.
- At the end of abstract, the authors can refer to future plan about DMTS's potential as a dietary supplement or adjunct therapy in terms of dosing, safety correlation to the human .
- Novelty over the following recently published article [ by the same authors ] should be highlighted in the introduction
Göntér, K., Dombi, Á., Kormos, V., Pintér, E., & Pozsgai, G. (2024). Examination of the Effect of Dimethyl Trisulfide in Acute Stress Mouse Model with the Potential Involvement of the TRPA1 Ion Channel. International Journal of Molecular Sciences, 25(14), 7701. https://doi.org/10.3390/ijms25147701
- In line 110, add country name.
- In line 126, correct it to (see section 2.5.1.)
- In line 263, write full name for PBS
- Check journal names in all references for example in refs 27, 77 journal name was missed.
- Statement in line 731 needs correction Current psychological and medicinal treatments of anxiety and depression focus largely on reducing negative mood rather than improving positive mood]. this statement is correct regarding old models but the field is shifting toward dual emphasis on reducing distress and enhancing positive functioning.
- Add future research plans
Author Response

(The authors gave the same response as above.)

Reviewer 4 Report
Comments and Suggestions for Authors
1. This manuscript presents interesting and novel findings on the role of dimethyl trisulfide (DMTS), a naturally occurring polysulfide, in modulating depression- and anxiety-like behaviors in a chronic stress model in mice, with a particular focus on the TRPA1 ion channel. The study is well-designed, uses appropriate methods, and presents comprehensive data. However, several major issues need to be addressed before the manuscript can be considered for publication.
2. While the introduction is thorough, the central hypothesis is not clearly and succinctly stated. Please revise the end of the introduction to explicitly state the study’s primary and secondary hypotheses.
3. The rationale for choosing DMTS over other polysulfides or TRPA1 agonists needs to be better justified.
4. The manuscript relies heavily on three-way ANOVA with complex interaction terms. While this is appropriate, the interpretation of these interactions needs to be clearer, especially in the behavioral and immunohistochemical sections. Provide more explanatory text in the results and discussion to clarify the biological relevance of significant interactions.
5. In some figures, significant effects are discussed without showing exact p-values or effect sizes (e.g., Figure 3 and Figure 4). Include these details to enhance transparency.
6. The selection of the 30 mg/kg dose is based on locomotor activity suppression, but the justification for its relevance to CNS effects or systemic exposure is insufficient. Include references or data that support this dose as pharmacologically relevant for CNS modulation.
7. Provide more discussion on DMTS pharmacokinetics and how its short half-life affects interpretation of repeated i.p. dosing.
8. The effect of repeated i.p. injections themselves (vehicle control) on stress and behavior needs further discussion. Could this have influenced the baseline behavior or hormonal response?
9. In the presentation of FOSB Immunohistochemistry results, The FOSB data are extensive and compelling but overwhelming in their current form. Consider summarizing key findings in a table and focus the text on the most relevant brain regions. Ensure consistency between micrographs and quantitative data.
10. Clarify whether FOSB expression reflects acute or cumulative activation across the CUMS period. Was any normalization to total cell counts done?
11. While TRPA1 KO mice are used, the specificity of DMTS for TRPA1 vs. other channels or receptors is not addressed. Could DMTS affect other TRP channels or CNS pathways? Please elaborate in the discussion.
12. Consider including (or suggesting future inclusion of) data using TRPA1 antagonists or pharmacological inhibition to strengthen the conclusion.
13. Improve the readability of figures by increasing font sizes and ensuring that all axes and labels are legible. Color coding or simplified legends would help distinguish treatment groups.
14. Include a summary figure or schematic diagram that illustrates the proposed mechanism of DMTS action via TRPA1.
15. Line 74: Clarify "more favourable pharmacokinetic properties" with specifics or citations.
16. IN the methodology, explicitly mention randomization and blinding procedures if used.
17. Reference formatting is inconsistent in some places; ensure adherence to journal style.
Abstract: "associated with protective functions such as antioxidant..." → consider rewording for conciseness. Several sections require language polishing for clarity and grammatical accuracy (e.g., abstract, introduction, results). Please revise the manuscript with the help of a native English speaker or professional editing service.
Author Response

(The authors gave the same response as above.)

Round 2
Reviewer 3 Report
Comments and Suggestions for Authors
The authors did all required recommendations. I appreciate their responses. The paper could be published in the current form.
Reviewer 4 Report
Comments and Suggestions for Authors
The queries raised are addressed.